# Facial Counterfactual Generation via Causal Mask-Guided Editing

**Pei-Sze Tan**                                                     *peiszetan26@gmail.com*
*CyPhi (ΨΦ) AI Lab, School of Information Technology*
*Monash University, Malaysia Campus*

**Sailaja Rajanala**                                        *sailaja.rajanala@monash.edu*
*CyPhi (ΨΦ) AI Lab, School of Information Technology*
*Monash University, Malaysia Campus*

**Arghya Pal**                                                  *arghya.pal@monash.edu*
*CyPhi (ΨΦ) AI Lab, School of Information Technology*
*Monash University, Malaysia Campus*

**Raphaël C.-W. Phan**                                     *raphael.phan@monash.edu*
*CyPhi (ΨΦ) AI Lab, School of Information Technology*
*Monash University, Malaysia Campus*
*&*
*Dept of Software Systems & Cybersecurity, Faculty of IT*
*Monash University, Clayton campus*

**Huey-Fang Ong**                                          *ong.hueyfang@monash.edu*
*CyPhi (ΨΦ) AI Lab, School of Information Technology*
*Monash University, Malaysia Campus*

**Reviewed on OpenReview:** *https://openreview.net/forum?id=ssamEGQjOC*

## Abstract

Generating counterfactual facial images is an important tool for interpretable machine learning, fairness analysis, and understanding the causal relationships among facial attributes. In this work, we propose a novel neuro-symbolic framework for *causal editing*, which integrates causal graph discovery, mask-guided counterfactual generation, and semantic interpretation to produce facial images that are both realistic and causally consistent. We first employ the Fast Causal Inference (FCI) algorithm to uncover latent causal relationships among facial attributes, enabling the identification of direct and indirect factors for target interventions. Using these causal graphs, we construct spatially informed masks that guide a DDPM-based generative model, ensuring that only regions relevant to the causal factors are modified. Finally, we leverage CLIP-based embeddings to provide logical, human-understandable explanations of the semantic changes in the counterfactuals. Experiments on CelebA and CelebA-HQ demonstrate that our approach produces high-fidelity counterfactuals, achieves superior performance on sparsity and realism metrics, and mitigates bias compared to state-of-the-art methods. This framework offers a principled approach to causally grounded, interpretable facial image editing. The code is available at: `https://github.com/noobasuna/FaCER`.

## 1 Introduction

Counterfactual reasoning has become an essential tool in interpretable and responsible machine learning (Velazquez et al., 2023). By considering how changes in inputs would alter outputs, counterfactuals allow researchers and practitioners to explore "what-if" scenarios in a controlled and understandable man-

ner (Goyal et al., 2019). In the context of facial images, the goal extends beyond explaining a model's decision: it involves generating realistic, causally consistent counterfactual instances that reflect plausible alternative appearances or attributes. Unlike traditional counterfactual explanation methods, which aim to minimally perturb inputs to reveal a model's decision boundary, counterfactual generation focuses on creating images that maintain fidelity to the underlying causal relationships among facial features.

Realism and causal consistency are critical in applications such as identity verification, biometric authentication, and human-computer interaction, as shown in Figure 1, an example in the visa application scenario and consequences of non-causal inference counterfactual generation. Existing image-based counterfactual methods often treat all features as independently manipulable, ignoring dependencies between attributes (Jeanneret et al., 2023; 2022). As a result, edits may produce implausible or entangled changes.

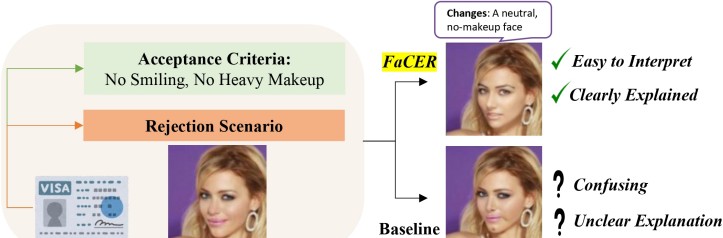

Figure 1: Motivation scenario: a rejected visa photo. Our method highlights the actual causal reasons, enhancing user understanding.

For example, altering hair colour might unintentionally modify perceived age or gender, and changing gender could introduce unrelated expressions. Such inconsistencies not only reduce trust in the generated images but can also amplify dataset biases, undermining fairness in downstream tasks (Xu et al., 2020).

Several methods have been proposed for generating visual counterfactual explanations. The work in (Goyal et al., 2019) developed a method for generating visual explanations with a counterfactual perspective. Another work that emphasises causal relation in counterfactual estimation is (Sanchez & Tsaftaris, 2022), a deep structural causal model that leverages recent advancements in diffusion models that involve iteratively sampling gradients from the marginal and conditional distributions that are part of the causal model. Peng et al. (Peng et al., 2022) simplified the explanation of model-detected face swap images by introducing automatic methods to create more authentic and manipulated counterfactual versions. Similarly, (Hendricks et al., 2018) introduced a phrase-critic model to enhance candidate explanations generated with inverted phrases.

However, methods that do not account for causal dependencies can yield visually unrealistic or non-actionable counterfactuals (Slack et al., 2021). De Sousa Ribeiro et al. (De Sousa Ribeiro et al., 2023) show that incorporating a causal model helps distinguish true causes from spurious associations, enabling accurate, high-fidelity counterfactual image generation. Similarly, Melistas et al. (Melistas et al., 2024) emphasise that realistic image edits must respect causal relationships inherent to the data generation process. These works demonstrate that causal models help isolate genuine causal factors and avoid spurious correlations. Existing methods face limitations that hinder the generation of causally consistent and interpretable facial counterfactuals. These gaps highlight the need for a framework that not only enforces causal consistency during counterfactual generation but also provides human-understandable explanations.

To address these challenges, we propose a framework that explicitly incorporates causal knowledge into the generation process. We introduce *Facial Counterfactual Generation via Causal Mask-Guided Editing*, a method that first learns a causal graph over facial attributes and perceptible features. This graph identifies direct causes and confounders, which then guide mask-based interventions during reverse diffusion. By modifying only causally relevant regions while holding confounders fixed, our approach ensures that generated counterfactuals are both realistic and semantically coherent. To further enhance interpretability, we leverage CLIP-based embeddings to provide textual descriptions of the visual edits, allowing users to understand which features were causally modified without implying an explanation of any specific model.

This work makes four primary contributions: (i) we shift the focus from counterfactual explanation to the causally coherent generation of facial images in a neuro-symbolic generation framework, emphasizing realistic and plausible outputs; (ii) we introduce a framework that learns causal structure over attributes

and applies mask-guided reverse diffusion to intervene selectively on direct causes; (iii) we provide visual and textual outputs that narrate the applied edits, improving interpretability and user understanding; and (iv) we demonstrate both quantitative and qualitative improvements in realism, sparsity, and causal consistency compared to state-of-the-art baselines on CelebA and CelebA-HQ.

## 2 Literature Review

There are two primary research gaps and challenges motivating this work: first, the difficulty of accurately discovering causal relationships among facial attributes in images; and second, the prevalence of stereotypical biases in current generative models.

**Causality in Facial Images Analysis**   The literature on causality in facial attributes has explored methods to improve model interpretability and achieve counterfactual fairness, particularly with respect to sensitive attributes such as race and gender. While prior work has addressed fairness by explicitly modeling the role of protected attributes (Mazumder & Singh, 2022), methods in facial attribute editing still face important limitations. For example, Kang et al. (Kang et al., 2022) propose generating synthetic facial replicas under factual and counterfactual assumptions using causal graph-based attribute translation, producing realistic counterfactual images. However, their work focuses on generating facial replicas for fairness evaluation rather than counterfactual explanations with emphasis on sparsity and causal consistency, which is a fundamentally different objective from ours. Other approaches incorporate wavelet scattering transforms (Liu et al., 2024) and channel attention for efficient face attribute classification, while NCINet (Wang & Boddeti, 2022) aims to identify causal relations from high-dimensional image representations.

Recent advances in diffusion-based counterfactual generation have made significant progress. Rasal et al. (Rasal et al., 2025) propose semantic abduction mechanisms for diffusion models, introducing spatial, semantic, and dynamic abduction to balance identity preservation with causal faithfulness during image editing. Wu et al. (Wu et al., 2025) present Variational Causal Inference (VCI), a framework that enables end-to-end counterfactual supervision during training with disentangled exogenous noise abduction. However, both approaches focus primarily on *image editing quality*—precise control over generated images and identity preservation—and assume *known causal graphs* as input. In contrast, our work addresses *interpretable counterfactual explanations* for fairness auditing and *learns the causal structure from data* via FCI, which is critical for real-world settings where ground-truth causal graphs are unavailable.

Despite these advances, existing methods face several limitations that hinder their practical application for causally consistent facial counterfactual generation. First, while some approaches build causal graphs to guide attribute changes, they rarely integrate these graphs directly into the image generation process, resulting in edits that may violate causal dependencies. Second, previous methods focus solely on visual output, without providing interpretable or human-understandable explanations of the applied interventions. Third, many causal discovery techniques rely on approximations that are sensitive to noise or missing variables, limiting robustness and scalability to large, high-dimensional facial features.

**Stereotyping in Generative Models**   Recent studies have highlighted significant concerns regarding stereotyping in image generation models, particularly in text-to-image systems. For instance, Bianchi et al. (Bianchi et al., 2023) demonstrated that widely accessible text-to-image models can amplify demographic stereotypes at large scale, producing biased images even from neutral prompts. Similarly, Barve et al. Barve et al. (2025) found that while prompt refinement can mitigate stereotypes, it often limits contextual alignment, indicating a trade-off between bias reduction and semantic accuracy. These issues underscore the need for more nuanced approaches to address bias in generative models.

However, existing literature predominantly focuses on text-to-image models, leaving a gap in understanding and mitigating bias in facial image generation. This oversight is particularly concerning given the widespread use of facial images in sensitive applications such as identity verification and biometric assessments. The lack of causal reasoning in current models often leads to unrealistic and biased facial edits, as they fail to account for the interdependencies among facial attributes.

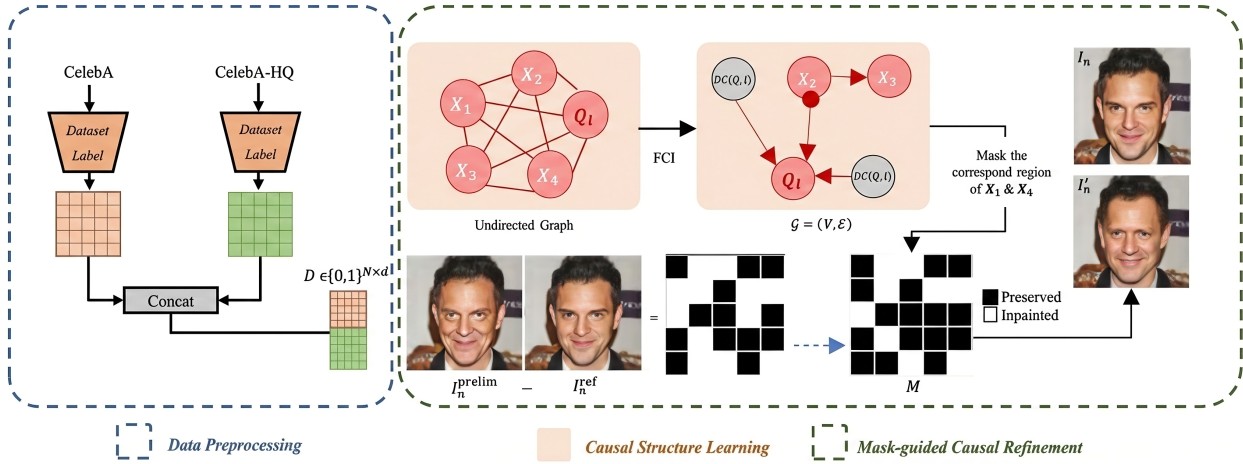

Figure 2: Our proposed causal editing pipeline. Data preprocessing constructs facial attribute matrix $D$, input to FCI. We assume $d = 5$, $N = 10$ for illustration. Using PAG $G$, we mask direct causes of target $Q_l$. The mask is obtained as binary difference between $I_n^{\text{ref}}$ and $I_n^{\text{prelim}}$, then filtered by causal structure to produce counterfactual $I_n'$. **Color coding:** red bold node = target attribute ($Q_l$); red nodes with white text = non-intervened attributes; grey nodes = direct causes $\text{DC}(Q_l)$ (inpainted regions). **Mask:** white = inpainted/regenerated regions; black = preserved regions.

## 3   Methodology

In Figure 2, it illustrates our complete pipeline. Starting from facial attribute matrix $D$, FCI discovers causal graph $G$. The reference and preliminary images are compared to construct mask $M$, which guides diffusion to produce the final counterfactual.

Table 1: Edge types in a Partial Ancestral Graph (PAG) and their interpretations.

| Edge Type | Interpretation |
|---|---|
| (a) $V_m \to V_n$ | Directed causal link from $V_m$ to $V_n$ (i.e., $V_n$ is a child of $V_m$). |
| (b) $V_m \circ\!\!\to V_n$ | Rules out $V_n$ as an ancestor of $V_m$; may reflect the presence of unmeasured confounders $C$. |
| (c) $V_m \circ\!\!-\!\!\circ V_n$ | Undetermined orientation; no $d$-separation (an unblocked path). |
| (d) $V_m \leftrightarrow V_n$ | Suggests an unobserved confounder between $V_m$ and $V_n$. |

### 3.1   Causal Graph Learning

To generate the causal graph, let $D \in \{0,1\}^{N \times d}$ be the data matrix of binary facial attributes, where each row corresponds to an instance and each column to an attribute. Given $D$, we use the Fast Causal Inference (FCI) algorithm Spirtes (2001) to estimate a Partial Ancestral Graph (PAG) over the $d$ attributes. Unlike conventional causal graphs, PAGs encode both directed causal edges and edges with uncertain orientation, as summarised in Table 1.

**Causal Graph Discovery.** To ensure causally consistent counterfactual generation, we first uncover the underlying causal structure among facial attributes. We compared established causal discovery algorithms, focusing on the FCI (Spirtes, 2001). While Peter-Clark (PC) (Spirtes et al., 2000) may suggest simplistic direct links, FCI uncovers richer, more nuanced structures that capture indirect effects and potential confounding relationships, allowing for more precise and valid interventions during counterfactual generation. Ultimately, we adopted FCI for its superior handling of hidden confounders—unobserved variables that affect multiple facial traits.

Mathematically, FCI employs chi-square tests to evaluate conditional dependencies between categorical variables:

$$\chi^2 = \sum \frac{(O - E)^2}{E}, \tag{1}$$

where $O$ is the observed frequency of a category and $E$ is the expected frequency under independence. Higher $\chi^2$ values indicate stronger dependence, with significance determined via comparison to a critical value or a p-value threshold of 0.05.

Our goal is to infer the causal structure among these attributes. Formally, we seek a graph $\mathcal{G} = (\mathcal{V}, \mathcal{E})$, where $\mathcal{V} = \{X_1, X_2, \ldots, X_d\}$ is the set of nodes and $\mathcal{E}$ is the set of edges encoding causal relations.

**Practical Advantages.** FCI combines conceptual robustness with computational efficiency. Its use of conditional independence tests avoids the heavy resource demands of deep learning-based causal discovery methods, making it scalable for large facial attribute datasets. For a complete mathematical treatment of FCI, see (Spirtes, 2001).

We provide extensive validation of the learned causal graph in Appendix A, including synthetic ground-truth recovery experiments, bootstrap stability analysis, and ablation studies demonstrating that causal knowledge—not graph topology alone—drives our performance improvements.

### 3.2 Reference Image Generation via Guided Diffusion

**Notation.** We use the following notation throughout this section: $I_n$ (original image), $I_n^{\mathrm{ref}}$ (reference counterfactual with full guidance), $I_n^{\mathrm{prelim}}$ (preliminary counterfactual with weak guidance), $I_n'$ (final counterfactual), $Q_l$ (target attribute), $\mathrm{DC}(Q_l)$ (direct causes of $Q_l$, identified as type (a) edges in the PAG), $M$ (binary mask), and $\lambda$ (mask threshold).

The first stage of our counterfactual generation produces a reference image that achieves the desired target attribute change without any causal constraints. Let $I_n$ denote the original image and $Q_l$ the target attribute. We use a guided DDPM to generate a reference counterfactual $I_n^{\mathrm{ref}}$:

$$x_{t-1} \sim \mathcal{N}\big(\mu_\theta(x_t, t) + \gamma \nabla_{x_t} \log p(Q_l | x_t), \sigma_t^2 I\big), \tag{2}$$

where $\mu_\theta(x_t, t)$ is the predicted mean from the diffusion model, $\sigma_t^2$ is the variance schedule, and $\gamma$ controls the guidance strength toward the target attribute $Q_l$.

This reference image serves as a minimally perturbed counterfactual that successfully flips the target label. At this stage, edits are guided purely by the attribute change, and no causal reasoning is applied. The image may include unrelated or unrealistic changes in other regions, which are addressed in the subsequent mask-guided refinement.

### 3.3 Mask-Guided Counterfactual Generation with Causal Graphs

After obtaining $I_n^{\mathrm{ref}}$, we identify regions relevant for causal intervention through a two-step process illustrated in Figure 2.

**Step 1: Generating Reference and Preliminary Images.**

1. **Reference image $I_n^{\mathbf{ref}}$:** Generated via guided diffusion toward the target attribute $Q_l$ with full guidance strength $\gamma$.

2. **Preliminary image $I_n^{\mathbf{prelim}}$:** Generated with weaker guidance to isolate attribute-specific changes, serving as a baseline for comparison.

**Step 2: Constructing the Causally-Filtered Mask.** The difference between these images highlights regions that changed due to the target attribute. We construct a binary mask $M$ that incorporates both the

pixel-level difference and the learned causal structure:

$$M_{i,j} = \begin{cases} 1, & \text{if } |I_n^{\text{ref}}(i,j) - I_n^{\text{prelim}}(i,j)| > \lambda \text{ and } (i,j) \in \text{DC}(Q_l) \\ 0, & \text{otherwise} \end{cases} \tag{3}$$

where $\lambda$ is a threshold controlling sensitivity to changes (see Appendix B for sensitivity analysis), and $\text{DC}(Q_l)$ denotes the spatial regions corresponding to *direct causes* of the target attribute $Q_l$. Direct causes are identified from the learned PAG as attributes connected to $Q_l$ via type (a) edges ($\rightarrow$), representing direct causal relations (Table 1).

**Step 3: Causal Filtering.** Using the PAG, we filter the mask to ensure only causally relevant regions are modified:

- **Direct cause regions** (type (a) edges: $\rightarrow$): Included in mask (white/inpainted)

- **Confounder or indirect cause regions** (bidirectional $\leftrightarrow$, circle endpoints $\circ\rightarrow$, $\circ\!-\!\circ$): Excluded from mask (black/preserved)

**Step 4: Counterfactual Generation.** The mask is applied within the DDPM generative process:

$$I_n' = G\big(\text{do}(I_n, M)\big), \tag{4}$$

where $G$ is the DDPM and $\text{do}(I_n, M)$ denotes intervening on $I_n$ along the causally relevant regions defined by $M$. This ensures that the final counterfactual $I_n'$ modifies only the direct causes of $Q_l$ while preserving confounded or unrelated attributes, resulting in both realistic and causally consistent counterfactual explanations.

**Scope of causal intervention.** We emphasise that our framework performs *causally-constrained editing* rather than formal pixel-level counterfactual inference. We conduct attribute-level do-intervention—identifying direct causal parents from the PAG—and use this to constrain spatial editing via the mask $M$. The ablation in Appendix A.4 confirms that specific causal relationships, not graph topology alone, drive our improvements.

### 3.4 Logical Interpretation

To provide a logical and human-understandable interpretation of the generated counterfactuals, we leverage CLIP (Radford et al., 2021) to map visual changes into semantic descriptions, as illustrated in Figure 3. Let $I_n$ denote the original image and $I_n'$ its counterfactual counterpart, and let $\mathcal{T}$ denote the space of textual descriptions. CLIP provides encoders $E_I$ and $E_T$ that embed images and texts into a shared $d$-dimensional space, enabling cross-modal reasoning.

Figure 3: Our proposed counterfactual-to-text interpretation model with CLIP-inspired image and text encoder.

We first compute the normalised image embeddings:

$$e = E_I(I_n), \quad e' = E_I(I_n'), \tag{5}$$

and define the *semantic transformation vector* as

$$\Delta e = e' - e. \tag{6}$$

This vector captures the overall change induced by the counterfactual transformation in the semantic embedding space.

To interpret these changes logically, we compare $\Delta e$ with a curated set of text embeddings $\{E_T(d_i)\}_{i=1}^n$, where each $d_i \in \mathcal{D} \subset \mathcal{T}$ represents an interpretable facial attribute or semantic concept. The relevance of each description is quantified using cosine similarity:

$$S(d_i) = \frac{\Delta e \cdot E_T(d_i)}{\|\Delta e\|_2 \cdot \|E_T(d_i)\|_2}. \tag{7}$$

Descriptions with high alignment are selected as the logical explanations for the visual changes:

$$\mathcal{D}^* = \{d_i \in \mathcal{D} \mid S(d_i) > \tau\}, \tag{8}$$

where $\tau$ is a similarity threshold.

This logical interpretation framework allows us to translate causal manipulations in the image into human-readable semantic statements, providing transparent insights into which facial attributes were modified and how these changes relate causally to the target attributes. By connecting visual counterfactuals to linguistic concepts, we bridge the gap between low-level image transformations and high-level reasoning about attribute causality.

## 4 Experiments

We conducted all our experiments on two large facial datasets to evaluate the performance of our model. Comparison is conducted with the current state-of-the-art counterfactual explanations model. Our experimental pipeline was implemented using PyTorch 2.0.1, with computations accelerated by NVIDIA A-100 GPUs on a SLURM-managed computing cluster using CUDA 11.7. The diffusion process is performed with 500 steps, with timestep respacing to 50, a stochastic sampling fraction of 0.1, and an inpainting ratio of 0.15. The causal discovery component of our work leveraged the `causal-learn` library (Zheng et al., 2023), a comprehensive Python package that implements both classical and cutting-edge causal discovery algorithms.

### 4.1 Datasets

Our work draws strength from the diversity and scale of the **CelebA** dataset (Liu et al., 2015), which is a collection featuring over 200,000 facial images with 40 precisely annotated attributes. To ensure our findings generalise across different image quality conditions, we also conducted experiments on **CelebA-HQ** (Lee et al., 2020), which is a high-resolution counterpart featuring 30,000 meticulously detailed images of 1,000 celebrities. These two datasets together provide a sufficiently diverse and large sample, covering a wide range of facial variations, attribute combinations, and image resolutions, making them well-suited for robust evaluation of our counterfactual image generation method. Additionally, CelebA and CelebA-HQ are standard benchmarks used by all baseline methods (ACE, DiME, STEEX, DiVE), enabling direct and fair comparison. We also conducted the attribute correlation and analysed the class imbalance of the CelebA dataset in Appendix C.

### 4.2 Evaluation Metrics

Our evaluation methodology employs four complementary metrics, each capturing different aspects of counterfactual quality.

*Realism of Generated Counterfactuals:* **Fréchet Inception Distance (FID)** serves as our primary gauge of generated image quality, with lower scores indicating better quality and closer resemblance to real images. Let $\mu_r, \Sigma_r$ be the mean and covariance of features from real images, and $\mu_g, \Sigma_g$ be the corresponding statistics for generated images. FID is computed as:

$$\text{FID} = \|\mu_r - \mu_g\|_2^2 + \text{Tr}\left(\Sigma_r + \Sigma_g - 2(\Sigma_r \Sigma_g)^{1/2}\right). \tag{9}$$

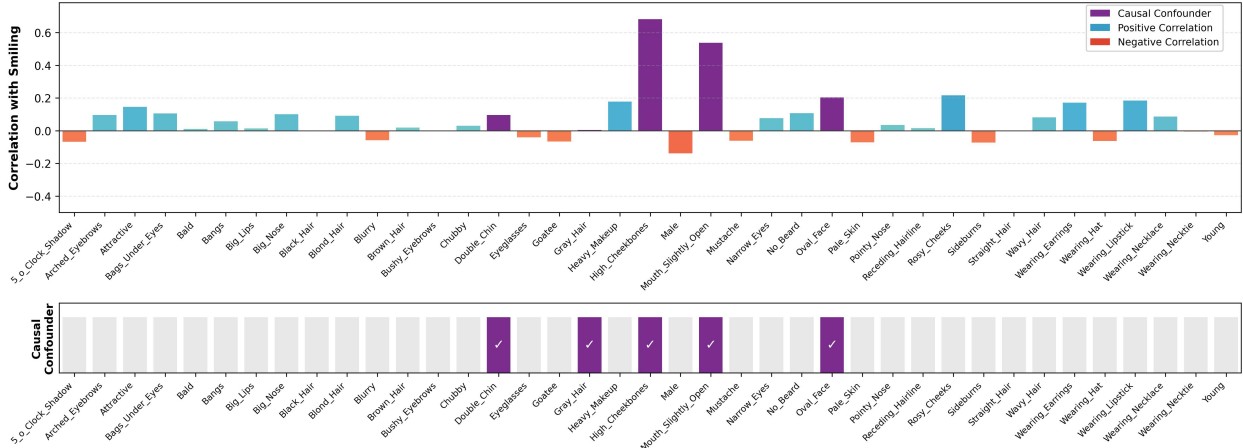

Figure 4: Correlations vs confounders in causal relationships of attribute "Smile" heatmap. The high purple bar in correlation ("High_cheekbones" and "Mouth_Slightly_Open") shows the conflict between highly correlated attributes and the confounders the causal algorithm identifies. The conflict is that while causal algorithms correctly identify confounders as indirect causes, traditional correlation-based models mistakenly learn these confounders as direct predictors simply because they are correlated with the target attribute.

We supplement this with **Scaled Fréchet Inception Distance (sFID)**, which adjusts the feature distributions to better capture differences across attribute scales:

$$\text{sFID} = \frac{1}{d} \sum_{k=1}^{d} \frac{(\mu_r^k - \mu_g^k)^2 + (\sigma_r^k - \sigma_g^k)^2}{(\sigma_r^k)^2 + \epsilon}, \tag{10}$$

where $d$ is the feature dimension, $\mu^k$ and $\sigma^k$ are the mean and standard deviation of the $k$-th feature, and $\epsilon$ is a small constant for numerical stability.

*Sparsity of Counterfactual Changes:* We introduce **Mean Number of Attributes Changed (MNAC)** to quantify how many facial attributes are modified in the counterfactual generation process. Let $x$ denote the original image and $\hat{x}$ the generated counterfactual, with corresponding binary attribute vectors $A(x)$ and $A(\hat{x})$. MNAC is defined as:

$$\text{MNAC} = \frac{1}{N} \sum_{i=1}^{N} \sum_{j=1}^{d} \mathbf{1}\big(A_j(x_i) \neq A_j(\hat{x}_i)\big), \tag{11}$$

where $N$ is the total number of test images, $d$ is the number of attributes, and $\mathbf{1}(\cdot)$ is the indicator function. Counterfactual explanations should exhibit minimal visual variations from the original image while achieving the desired change in target attributes. Lower MNAC indicates sparser, more targeted modifications—a desirable property for interpretable counterfactual explanations.

*Identity Preservation:* To ensure that counterfactual edits preserve the identity of the subject, we employ **Face Verification Accuracy (FVA)** Cao et al. (2018); Jeanneret et al. (2023). FVA measures whether the original image and its counterfactual are recognised as the same individual by a face verification network. Specifically, we extract deep embeddings using ArcFace (Deng et al., 2019) and compute the cosine similarity between the original and counterfactual image embeddings. FVA is defined as:

$$\text{FVA} = \frac{1}{N} \sum_{i=1}^{N} \mathbf{1}\Big( \cos\big(f(x_i), f(\hat{x}_i)\big) > \tau \Big), \tag{12}$$

where $f(\cdot)$ denotes the ArcFace embedding function, $\cos(\cdot, \cdot)$ is the cosine similarity, and $\tau$ is the verification threshold. Higher FVA indicates better identity preservation. This metric is crucial for counterfactual explanations, as meaningful explanations should modify only the target attribute while keeping the subject's identity intact.

Notably, we deliberately avoid using correlation-based metrics such as Correlation Difference (CD) because our method is specifically designed to account for causal relationships rather than mere correlations between facial attributes, as illustrated in Figure 4.

### 4.3 Baseline Methods

We compared our model with several state-of-the-art baselines for counterfactual explanation generation. **Adversarial Counterfactual visual Explanations (ACE)** (Jeanneret et al., 2023) leverages adversarial attacks on a diffusion model to refine counterfactual images, focusing on generating subtle yet effective modifications that change the target attribute while maintaining realism. **Steering Counterfactual Explanations with Semantics (STEEX)** Jacob et al. (2022) generates counterfactuals by steering the latent representations of images in a pretrained generative model, guided by semantic attribute changes, allowing controlled modifications of specific attributes while preserving other visual details. **Diffusion Model Counterfactual Explanations (DiME)** Jeanneret et al. (2022) uses diffusion-based generative models to produce realistic counterfactual images through iterative denoising of latent representations conditioned on target attributes, enabling smooth transitions between original and modified attribute states. **Counterfactual Explanations with Diverse Valuable Explanations (DiVE)** Rodríguez et al. (2021) focuses on generating a diverse set of counterfactual explanations that are both visually plausible and informative, employing a diversity-promoting objective to ensure multiple counterfactuals for a given input capture different plausible modifications.

These baselines cover a wide spectrum of counterfactual generation paradigms, from adversarial refinement and latent-space steering to diffusion-based synthesis and diversity-driven explanations, making them suitable benchmarks for evaluating the performance of our proposed method.

We restrict comparison to methods that (i) generate image-level facial counterfactuals, (ii) operate on attribute-conditioned diffusion or generative models, and (iii) preserve identity in facial editing tasks. While other causal generative frameworks exist (e.g., diffusion-based causal models evaluated on non-facial datasets such as BDD or ImageNet), they either assume known causal graphs, do not focus on facial attribute editing, or require retraining under different supervision regimes. Our selected baselines therefore represent the most directly comparable methods for facial counterfactual generation under the same evaluation setting.

### 4.4 Results and Discussions

We presented and discussed both the causal graph results as illustrated in Figure 5, 6 and the generated counterfactual explanations images quantitative and qualitative evaluation are concluded in Table 2 and 3.

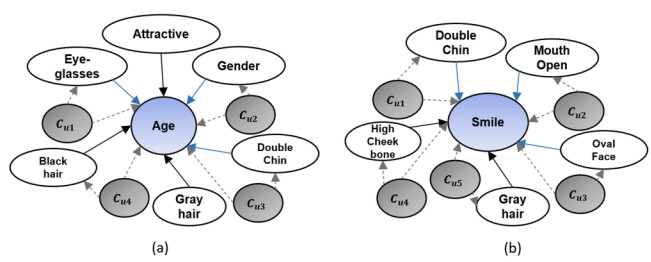 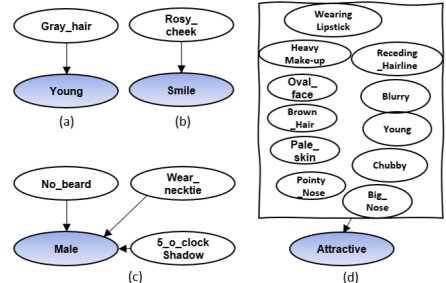

Figure 5: Partial Ancestral Graph (PAG) for target attributes "Young" and "Smiling". Blue: target attribute $Q_l$; grey: confounders $C$; white: related attributes $I$. Bidirectional edges ($\leftrightarrow$) indicate unobserved confounders (e.g., Smiling $\leftrightarrow$ Mouth_Open implies Smiling $\leftarrow C_u \rightarrow$ Mouth_Open). See Table 1 for edge interpretations.

Figure 6: CelebA dataset causal relation of the target variable (in blue) with the PC algorithm. Note that the label in this graph indicates the attribute class type without specifying its actual binary class value in CelebA.

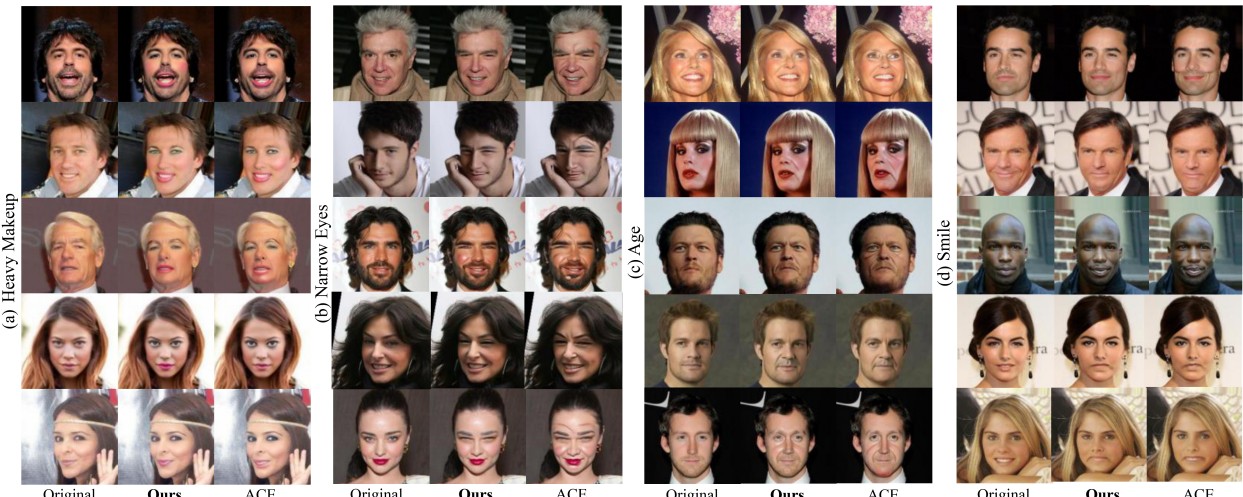

Figure 7: Qualitative counterfactuals results for facial attributes demonstrate the effectiveness of our approach. For "Age", the baseline often generates features correlated with aging, such as eye bags and glasses. However, our causal graph analysis from Figure 5 indicates that "Bags Under Eyes" does not directly influence "Age". For "Smile", our method successfully disentangles correlated attributes like "High Cheekbones" and "Opening Mouth", providing more accurate and realistic counterfactual explanations. For additional examples and higher-resolution images, please refer to the supplementary material. Zoom for best view.

Table 2: Quantitative results for CelebA and CelebAHQ datasets.

| | CelebA | | | | | | | | CelebAHQ | | | | | | | |
| | Smile | | | | Age | | | | Smile | | | | Age | | | |
| Method | FID↓ | sFID↓ | MNAC↓ | FVA↑ | FID↓ | sFID↓ | MNAC↓ | FVA↑ | FID↓ | sFID↓ | MNAC↓ | FVA↑ | FID↓ | sFID↓ | MNAC↓ | FVA↑ |
|---|---|---|---|---|---|---|---|---|---|---|---|---|---|---|---|---|
| STEEX (Jacob et al., 2022) | 10.2 | - | 4.11 | 96.9 | 11.8 | - | 3.44 | 97.5 | 21.9 | - | 5.27 | 97.6 | 26.8 | - | 5.63 | 96.0 |
| DiVE (Rodríguez et al., 2021) | 29.4 | - | - | 97.3 | 33.8 | - | 4.58 | 98.2 | 107.0 | - | 7.41 | 35.7 | 107.5 | - | 6.76 | 32.3 |
| DiME (Jeanneret et al., 2022) | 3.17 | 4.89 | 3.72 | 98.3 | 4.15 | 5.89 | 3.13 | 95.3 | 18.1 | 27.7 | 2.63 | 96.7 | 18.7 | 27.8 | 2.10 | 95.0 |
| ACE $\ell_1$ (Jeanneret et al., 2023) | 1.27 | 3.97 | 2.94 | 99.9 | 1.45 | 4.12 | 3.20 | 99.6 | 3.21 | 20.2 | 1.56 | 100.0 | 5.31 | 21.7 | 1.53 | 99.6 |
| ACE $\ell_2$ (Jeanneret et al., 2023) | 1.90 | 4.56 | 2.77 | 99.9 | 2.08 | 4.62 | 2.94 | 99.6 | 6.93 | 22.0 | 1.87 | 100.0 | 16.4 | 28.2 | 1.92 | 99.6 |
| **Ours** | **1.08** | **3.79** | **1.63** | **99.9** | 1.51 | **4.10** | **1.92** | **99.8** | **1.27** | **5.12** | **1.55** | 99.9 | 12.28 | **18.83** | **1.06** | **99.9** |

### 4.4.1 Causal Graph Analysis

The learned causal structures are illustrated in Figure 5 (PAG) and Figure 6 (DAG obtained via the PC algorithm). For the query labels "Young" and "Smile" as child nodes: For "Young", the graph includes a direct edge from "Gray Hair" to "Young", while "Eyeglasses", "Black Hair" and "Double Chin" appear as indirect (upstream) factors. Because "Black Hair" and "Gray Hair" are mutually exclusive and may be confounded with age-related appearance, we avoid direct interventions on these attributes when orientation is uncertain. Accordingly, our masking targets regions associated with "Eyeglasses" and "Double Chin". For "Smile", the graph indicates upstream influences from "Double Chin", "High Cheekbones", and "Gray Hair". We therefore apply masks to the corresponding regions when guiding the generation process, while avoiding edits along edges marked as uncertain or confounded in the PAG.

### 4.4.2 Quantitative Results

As shown in Table 2, our model consistently outperforms prior methods across both CelebA and CelebAHQ datasets. It achieves the lowest FID and sFID scores in most cases, especially for the "smile" attribute, indicating superior image quality and more realistic feature preservation. Moreover, ours obtains significantly better MNAC scores, highlighting enhanced diversity and causal disentanglement in the generated images. Our analysis reveals that the classifier often relies on spurious correlations learned during training rather than causal relationships, resulting in lower MNAC values for these attributes, a limitation also acknowledged by DiME (Jeanneret et al., 2022).

Table 3: Quantitative Assessment for CelebA. H. Makeup and N. Eyes are heavy makeup and narrow eyes attributes.

| Attr. | Gender | H. Makeup | N. Eyes |
|-------|--------|-----------|---------|
| MNAC (↓) | | | |
| ACE | 3.95 | 5.18 | 3.44 |
| **Ours** | **3.91** | **5.18** | **2.84** |
| sFID (↓) | | | |
| ACE | 7.56 | 11.29 | **4.04** |
| **Ours** | **7.42** | **10.98** | 4.44 |

Table 4: Quantitative bias metrics. Higher DI and lower SPD indicate reduced bias.

| CelebA | | DI (↑) | SPD (↓) |
|--------|-----|--------|---------|
| Age-HighCheek | ACE | 2.37 | 0.019 |
| | **Ours** | **2.52** | **0.011** |
| Age-H.Makeup | ACE | 6.23 | 0.159 |
| | **Ours** | **6.24** | **0.152** |
| Gender-H.Makeup | ACE | 0.019 | 0.016 |
| | **Ours** | **0.020** | **0.015** |

### 4.4.3 Qualitative Results

Figure 7 (more in Appendix D) illustrates this, showing how the baseline's counterfactuals for "young" often include ageing cues like eye bags and glasses. Similarly, confounding influences are evident for "smile", with high cheekbones and open mouths being incorrectly emphasised. In contrast, our ageing effects appear more realistic. Also, ours generates images with more noticeable heavy makeup than the baseline. In contrast, our approach successfully corrected the classification by generating impactful counterfactuals. Our method integrates causal knowledge to produce realistic, less biased counterfactual images, revealing the classifier's decision process. This enhances interpretability and utility in real-world applications where human understanding is crucial. For example, the lines generated by baseline ACE in Figure 7 for age may be confusing to the general public, as they represent ageing effects.

We present qualitative results for the "Heavy Makeup" and "Narrow Eyes" attributes in Figure 7 (a) and (b), with additional bias evaluations in the supplementary materials. These attributes, often linked to sensitive factors like gender and race, are not explicitly labeled in the dataset. Although not considered in prior work, we find them important for uncovering potential biases and enhancing the fairness of counterfactual explanations.

### 4.4.4 Fairness Evaluation

Table 4 presents bias metrics for different attribute pairs evaluated on the CelebA dataset. We used a ResNeXt-50 classifier trained for 35 epochs to validate attribute intensity in generated counterfactual images. Each row examines a specific attribute pair where the first attribute represents the protected group (e.g., Age, Gender) and the second denotes the unprivileged group (e.g., HighCheek, H.Makeup).

We evaluate fairness using two widely-adopted metrics: **Disparate Impact (DI)** and **Statistical Parity Difference (SPD)**.

**Disparate Impact (DI):** DI measures the ratio of positive outcomes between the unprivileged and protected groups. It is defined as:

$$\text{DI} = \frac{\Pr(\hat{Y} = 1 \mid A = \text{unprivileged})}{\Pr(\hat{Y} = 1 \mid A = \text{protected})}, \tag{13}$$

where $\hat{Y}$ denotes the predicted attribute, and $A$ represents the group membership. A DI value closer to 1 indicates equitable treatment, with higher values suggesting reduced bias against the unprivileged group.

**Statistical Parity Difference (SPD):** SPD quantifies the difference in positive prediction rates between the protected and unprivileged groups:

$$\text{SPD} = \Pr(\hat{Y} = 1 \mid A = \text{protected}) - \Pr(\hat{Y} = 1 \mid A = \text{unprivileged}). \tag{14}$$

Lower absolute values of SPD indicate smaller disparities, with SPD = 0 representing perfect statistical parity.

We compare against ACE as it achieves the strongest performance on realism metrics among all baselines (Table 2). Other baselines such as DiME and STEEX exhibit substantially weaker image quality (FID

> 3 for DiME, > 10 for STEEX), and evaluating fairness on lower-quality counterfactuals would conflate generation quality issues with fairness metrics, making interpretation difficult.

**Connection to Stereotype Bias Reduction:** Stereotypes arise when spurious correlations are treated as causal relationships. Our method reduces stereotype bias through three mechanisms:

1. **Causal disentanglement:** The learned PAG identifies direct causes versus spurious correlations. For example, in Figure 7 ("age" column), baselines add "Double Chin" when aging (a correlated but not causal attribute), while our PAG shows Double_Chin $\leftrightarrow$ Age indicates confounding—so we exclude this region from modification.

2. **Mask-guided intervention:** We only modify regions corresponding to direct causal parents. If "Smiling" and "Female" are correlated but not causally related, our mask excludes gender-associated regions, preventing inadvertent feminization (see Appendix E).

3. **Quantitative evidence:** Table 4 shows improved DI (closer to 1.0) and SPD (closer to 0.0) compared to baselines.

By filtering correlation from causation—only direct causal edges ($\rightarrow$) are included while all other edge types are excluded—our PAG prevents stereotyped attributes from propagating into counterfactuals.

**Metrics for Measuring Stereotype Bias:** DI and SPD directly quantify group fairness disparities. Additionally, MNAC (Mean Number of Attribute Changes) indirectly measures stereotype bias: lower MNAC indicates the model does not automatically add stereotypically correlated attributes when modifying the target attribute. Our method achieves MNAC of 1.63 compared to 2.94 for baselines without causal constraints.

Our method consistently achieves lower SPD values and comparable or higher DI values across all evaluated attribute pairs compared to the ACE baseline. This demonstrates that our approach effectively mitigates bias by reducing prediction disparities between different demographic groups while maintaining equitable performance across attributes in counterfactual explanations. The improvement stems from our causal disentanglement: by learning the PAG and restricting edits to direct causal parents only, we prevent spurious correlations (which often encode societal biases) from propagating into the generated counterfactuals. In other words, generated counterfactuals using our method exhibit a fairer distribution of attribute changes, ensuring that the protected and unprivileged groups are treated more equitably.

### 4.4.5 Logical Interpretation

To quantitatively assess the semantic alignment between images and their corresponding text descriptions, we report two evaluation metrics (Radford et al., 2021; Faghri et al., 2018): *Recall@1* (R@1) and *mean similarity*. R@1 measures whether the intended attribute change is the top-detected semantic change, reflecting how precisely the counterfactual targets the desired modification. Mean similarity denotes the average CLIP similarity score between each image and its descriptive caption, measuring overall semantic alignment (see example in Appendix F).

Table 5: CLIP-based semantic alignment evaluation comparing FaCER with SOTA baseline.

| Category | Method | Mean Sim. | R@1 (%) |
|---|---|---|---|
| Smile | ACE | 6.22 | 49.09 |
| | FaCER (Ours) | 5.91 | 72.95 |
| Age | ACE | 4.45 | 62.98 |
| | FaCER (Ours) | 5.03 | 61.34 |

We use CLIP (Radford et al., 2021) as our vision-language model due to its open-source availability, computational efficiency, and status as the standard embedding space for vision-language alignment tasks. Unlike prior work that applies CLIP-based interpretation to arbitrary attribute changes, our approach grounds these explanations in the learned causal structure—only direct causes identified by FCI are modified, ensuring that semantic interpretations reflect causally meaningful edits rather than spurious correlations.

Table 5 compares FaCER against ACE, the strongest baseline on realism metrics. For Smile, FaCER achieves substantially higher R@1 (72.95% vs. 49.09%), indicating that our counterfactuals more precisely target the intended attribute as the primary semantic change. For Age, results are comparable (61.34% vs. 62.98%). Notably, for Smile, FaCER achieves higher R@1 despite a lower mean similarity score (5.91 vs. 6.22). This combination suggests that our method produces more *focused, attribute-specific* modifications: the counterfactual changes are concentrated on the target attribute rather than introducing broader modifications that may inflate similarity scores while affecting unintended attributes. This behaviour aligns with our goal of generating minimal, interpretable counterfactual explanations—edits should change only what is causally necessary.

## 5    Conclusion

We introduced a framework for causal editing of facial images that unifies causal discovery, mask-guided counterfactual generation, and semantic interpretation. Motivated by the need for interpretable and causally grounded visual reasoning, our approach employs the Fast Causal Inference (FCI) algorithm to uncover relationships among facial attributes and uses spatially informed masks to guide a diffusion-based generator, ensuring that only causally relevant regions are modified. Experiments on CelebA and CelebA-HQ show that our method produces realistic, identity-preserving counterfactuals and surpasses state-of-the-art baselines on sFID and MNAC metrics. The integration of CLIP-based semantic explanations further enhances interpretability and supports fairness analysis. While our framework assumes reasonably accurate causal graphs, which may not always hold in noisy real-world settings, it provides a principled foundation for interpretable generative modelling. Future work will address uncertain causal structures and feature-level alternatives, and extend our framework to broader domains such as medical imaging and scene understanding.

### Broader Impact Statement

Our work introduces a causal mask-guided framework for generating counterfactual facial images to improve interpretability and fairness evaluation in machine learning. While such methods can help reveal causal relationships and reduce reliance on spurious correlations, techniques for editing facial images may also pose risks of misuse, such as generating misleading or manipulated visual content. Additionally, biases in the underlying datasets may still influence generated counterfactuals. Therefore, this work is intended primarily for research on interpretable and fair machine learning, and its use should consider appropriate safeguards and responsible deployment.

### Acknowledgement

This work was supported in part by the Uxpress Project, funded by the Prader-Willi Syndrome Association Victoria & Monash Faculty of Information Technology, Australia, and by the Æinstein Project funded by Petronas Research.

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

# Appendix

## A   Causal Graph Validation

We provide additional validation experiments for the learned causal graph as requested by reviewers.

### A.1   Synthetic Ground-Truth Recovery

To verify that our causal discovery pipeline correctly identifies known causal structures, we constructed a synthetic Structural Causal Model (SCM) with ground-truth edges. Both PC and FCI algorithms correctly recover the adjacency structure (precision = 1.0, recall = 1.0), confirming that our implementation faithfully identifies causal relationships when the true structure is known.

### A.2   Bootstrap Stability Analysis

To assess the stability of learned causal structures, we performed 500 bootstrap runs on CelebA. Table 6 reports the frequency with which each candidate parent appears in the learned graph for selected target attributes.

Table 6: Bootstrap stability analysis over 500 runs on CelebA. Frequencies indicate how often each candidate parent appears in the learned causal graph.

| Target | Candidate Parents | Bootstrap Freq. |
|--------|-------------------|-----------------|
| Young | Gray_Hair | 28% |
| | Bald | 22% |
| | Attractive | 14% |
| Smiling | High_Cheekbones | 30% |

The moderate frequencies reflect genuine structural uncertainty due to latent confounding—precisely what motivates our use of Partial Ancestral Graphs (PAGs) learned from FCI rather than Directed Acyclic Graphs (DAGs) learned from algorithms such as PC. As shown in Figure 5 of main paper, relations such as Smile $\leftarrow U_2 \rightarrow$ Mouth_Open indicate latent confounders that make parent sets only partially identifiable. This is not a limitation but an accurate representation of causal ambiguity inherent in observational data.

Importantly, implausible parents (e.g., Wearing_Hat $\rightarrow$ Smiling) never appear in any bootstrap run, confirming that our approach appropriately filters spurious relationships.

### A.3   Algorithm Sensitivity

We evaluated sensitivity to the choice of causal discovery algorithm and hyperparameters:

- **Algorithm comparison:** FCI and PC show high structural agreement (Jaccard similarity = 0.95).
- **CI-test sensitivity:** Results are stable across different conditional independence tests ($\chi^2$, $G^2$).
- **Significance level:** Structure remains stable across $\alpha \in \{0.01, 0.05, 0.1\}$.

This consistency confirms that our findings are not artifacts of specific algorithmic choices.

### A.4   Ablation: Impact of Causal Graph

To isolate the contribution of causal knowledge, we compare our full method against a version without the causal graph (equivalent to ACE with $\ell_1$ regularization).

Without the causal graph, our method reduces to baseline performance. The substantial increase in MNAC ($1.63 \rightarrow 2.94$) demonstrates that causal knowledge—not masking alone—drives our improvements in attribute

Table 7: Ablation study on the contribution of the causal graph.

| Configuration | FID ↓ | MNAC ↓ |
|---|---|---|
| With causal graph (Ours) | **1.08** | **1.63** |
| Without causal graph (ACE $\ell_1$) | 1.27 | 2.94 |

specificity. This ablation directly confirms that the learned causal structure is essential for generating minimal, targeted counterfactual explanations.

## B Hyperparameter Sensitivity Analysis

### B.1 Mask Threshold $\lambda$

The mask threshold $\lambda$ controls which pixels are included in the editable region based on the difference between the reference and preliminary images (Equation 3 in main paper). We set $\lambda = 0.1$ (normalized pixel intensity threshold) based on the sensitivity analysis presented in Table 8.

Table 8: Sensitivity analysis for mask threshold $\lambda$ on CelebA (Smile attribute). Lower $\lambda$ produces more focused masks, while higher $\lambda$ leads to excessive modifications.

| $\lambda$ | FID ↓ | MNAC ↓ |
|---|---|---|
| **0.10** | **1.08** | **1.63** |
| 0.50 | 1.21 | 2.32 |
| 1.00 | 1.68 | 3.63 |

As shown in Table 8, increasing $\lambda$ leads to progressively worse performance on both metrics. With $\lambda = 1.0$, MNAC increases from 1.63 to 3.63, indicating that the model modifies more than twice as many attributes as necessary. This occurs because a higher threshold allows more pixels to be included in the mask, leading to unnecessary modifications in regions unrelated to the target attribute. The degradation in FID ($1.08 \rightarrow 1.68$) suggests that these excessive edits also harm image realism.

We select $\lambda = 0.1$ as it achieves the best balance between targeted attribute modification (lowest MNAC) and image quality (lowest FID). This value ensures that only pixels with substantial differences between the reference and preliminary images are included in the mask, effectively isolating the regions most relevant to the target attribute change.

### B.2 Complementary Relationship between FID and sFID

We use both FID and sFID as complementary metrics for evaluating counterfactual realism, as they capture different aspects of image quality.

**FID (Fréchet Inception Distance)** compares the full feature distributions between real and generated images, including feature covariances. This makes FID sensitive to incoherent attribute combinations—if a generated image contains an implausible combination of features (e.g., young face with gray hair), FID will detect this as a deviation from the real data distribution. FID thus provides a holistic measure of overall image realism and distributional faithfulness.

**sFID (Scaled Fréchet Inception Distance)** complements FID by measuring per-feature realism without considering cross-feature covariances. This is useful for assessing whether individual attributes are rendered realistically, independent of their combinations. sFID can reveal issues in specific feature dimensions that might be masked by the aggregate FID score.

The two metrics serve complementary purposes:

- FID captures *global coherence*—whether the overall image and attribute combinations are realistic and in-distribution.

- sFID captures *local fidelity*—whether individual features are rendered with high quality.

By reporting both metrics, we provide a more complete picture of counterfactual quality. A method with good FID but poor sFID might produce globally coherent but locally degraded images, while a method with good sFID but poor FID might render individual features well but combine them in implausible ways. Our method achieves strong performance on both metrics, indicating that the generated counterfactuals are both globally coherent and locally realistic.

## C  Understanding Attribute Correlations

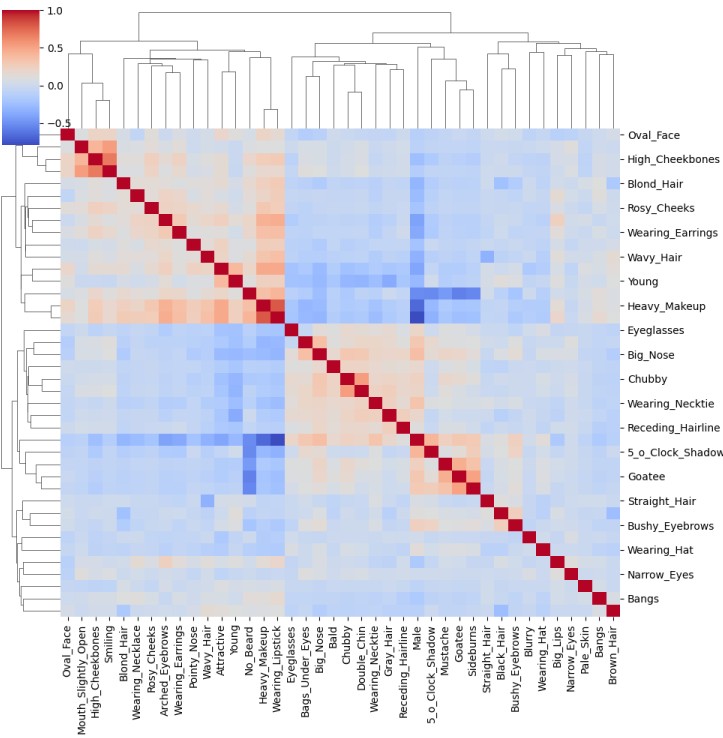

Figure 8: Dendrogram and correlation heatmap for the CelebA attribute classes. The intensity of color (blue or red) indicates the strength of correlation between attribute pairs. Hierarchical clustering shows which attributes frequently co-occur in the dataset.

As shown in Fig. 8, we visualize the correlation patterns between different facial attributes in CelebA using a hierarchical clustering dendrogram and heatmap. The dendrogram reveals how attributes cluster based on correlation patterns, with shorter branches indicating higher similarity. However, it is crucial to emphasize that these correlations do not imply causation.

For example, consider two variables $X$ and $Y$ with correlation coefficient $r = \frac{\text{cov}(X,Y)}{\sigma_X \sigma_Y}$. Despite a strong correlation value, this does not necessarily mean that changes in $X$ cause changes in $Y$, or vice versa. Other factors, including confounding variables, may be responsible for the observed relationships.

In our analysis, we found that existing methods consistently associate "smiling" with high cheekbones and open mouth due to dataset correlations, limiting the diversity of generated images. Our causal relationship-aware model effectively mitigates such biases, producing more realistic facial attribute manipulations.

## C.1   Addressing Class Imbalance in CelebA

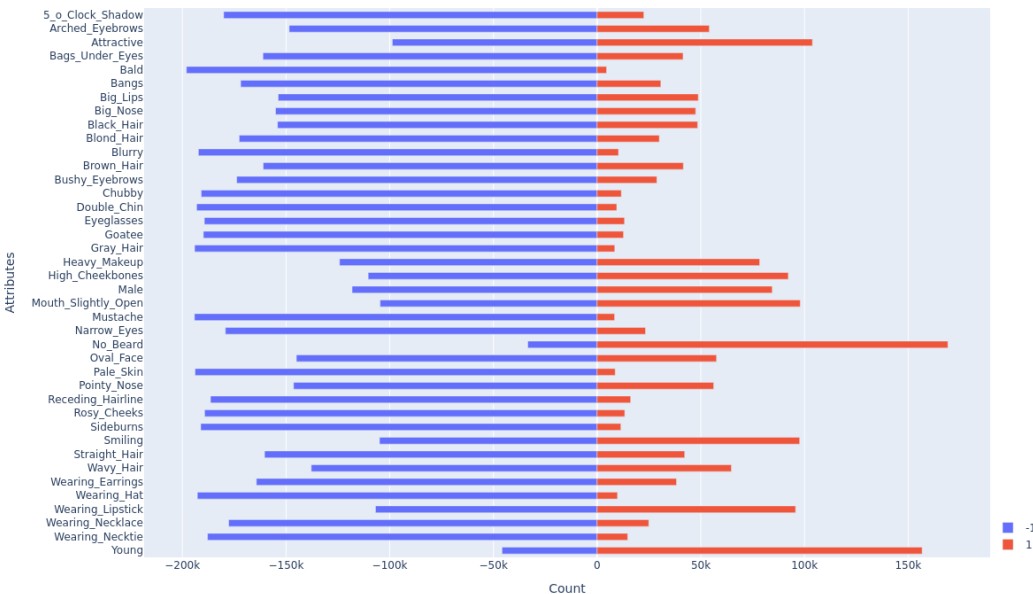

Figure 9: Class distribution for facial attributes in the CelebA dataset, highlighting significant imbalances across many attribute categories.

The CelebA dataset exhibits substantial class imbalance across different attributes, as shown in Fig. 9. This visualization highlights how attributes such as "Smiling" have disproportionate sample distributions between positive and negative classes. Such imbalances can significantly impact model performance, particularly when generating images with underrepresented attributes. Our method addresses this challenge by incorporating causal relationships to generate diverse images with balanced attribute intensities, improving both quality and diversity.

**Addressing Perceptual Interpretability** While counterfactual explanations are crucial for understanding model behavior, their effectiveness hinges on their interpretability to humans. Existing methods often struggle in this regard, with high flip rates resulting in imperceptible or nonsensical changes to human observers.

Our approach focuses on enhancing the perceptual interpretability of counterfactual explanations by prioritizing changes that are more noticeable and coherent to humans. FaCER identifies and emphasizes features salient to human perception, such as altering the hairstyle or adding/removing glasses. By doing so, we ensure that the counterfactual images are technically valid, visually compelling, and aligned with human expectations.

## D   Additional Evaluation Results

We compare our method with the state-of-the-art approach Jeanneret et al. (2023) on both CelebA and CelebAHQ datasets. The results in Figures 10, 11, 12, and 13 demonstrate that our method generates more realistic images with diverse attributes such as *smiling* and *young*. The difference maps highlight how our method makes more targeted changes to relevant facial features while preserving unrelated attributes. Our approach effectively captures the attribute intensity and diversity in the generated images, enhancing both realism and quality compared to existing methods.

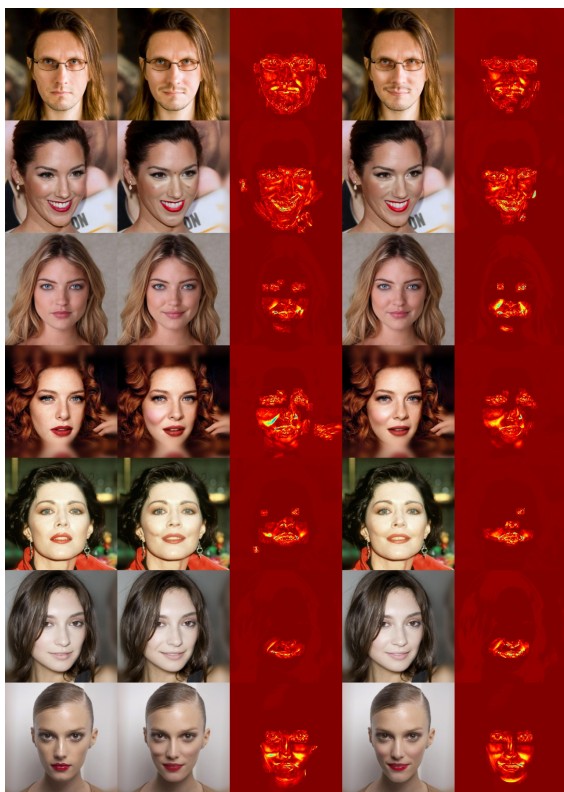

Figure 10: Qualitative comparison on CelebAHQ for the *smiling* attribute. From left to right: (1) Original image, (2) Baseline method result, (3) Difference map between baseline and original, (4) Our method result, (5) Difference map between our method and original. Note how our method generates more realistic attribute manipulations with fewer unintended changes to other facial features.

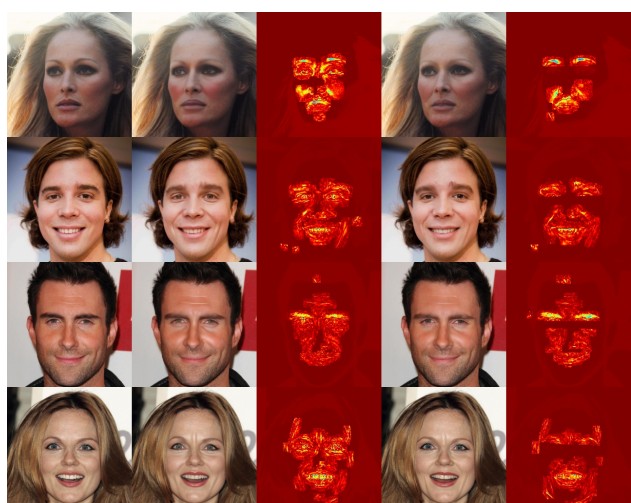

Figure 11: Qualitative comparison on CelebAHQ for the *young* attribute. From left to right: (1) Original image, (2) Baseline method result, (3) Difference map between baseline and original, (4) Our method result, (5) Difference map between our method and original. Our method better preserves identity features while realistically altering age appearance.

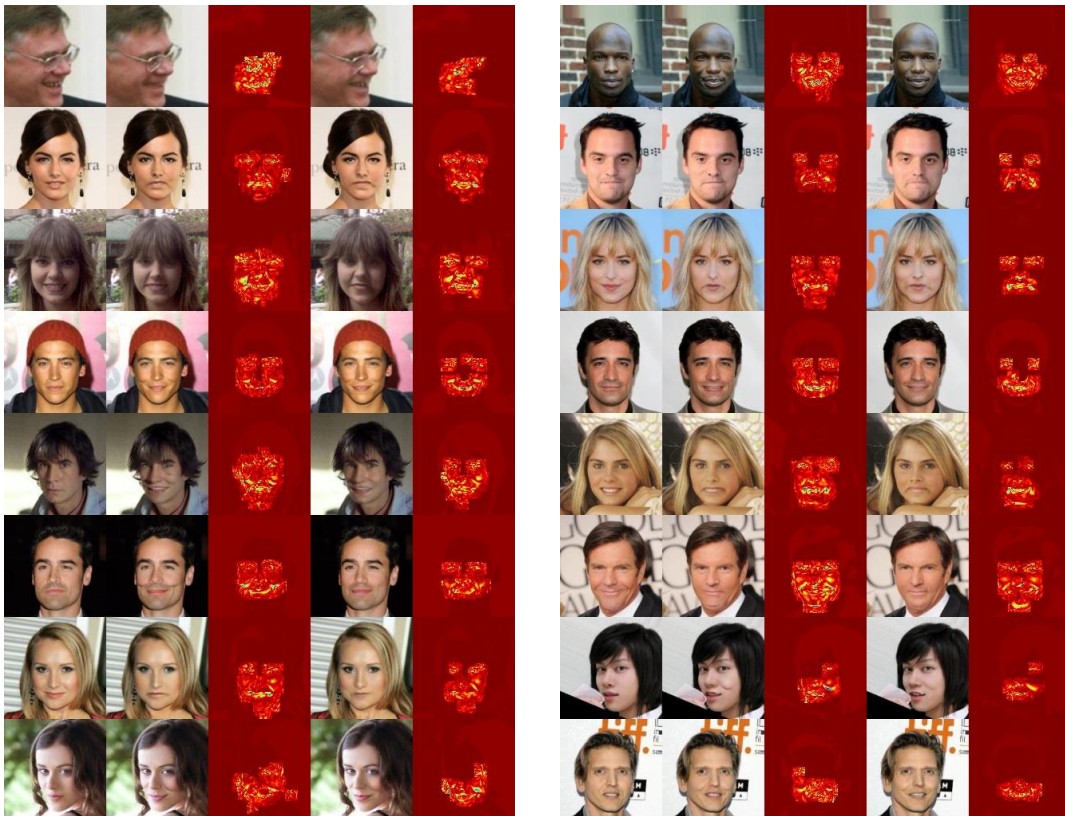

Figure 12: Qualitative comparison on CelebA for the *smiling* attribute. From left to right: (1) Original image, (2) Baseline method result, (3) Difference map between baseline and original, (4) Our method result, (5) Difference map between our method and original. Our approach generates more natural smiles with fewer artifacts.

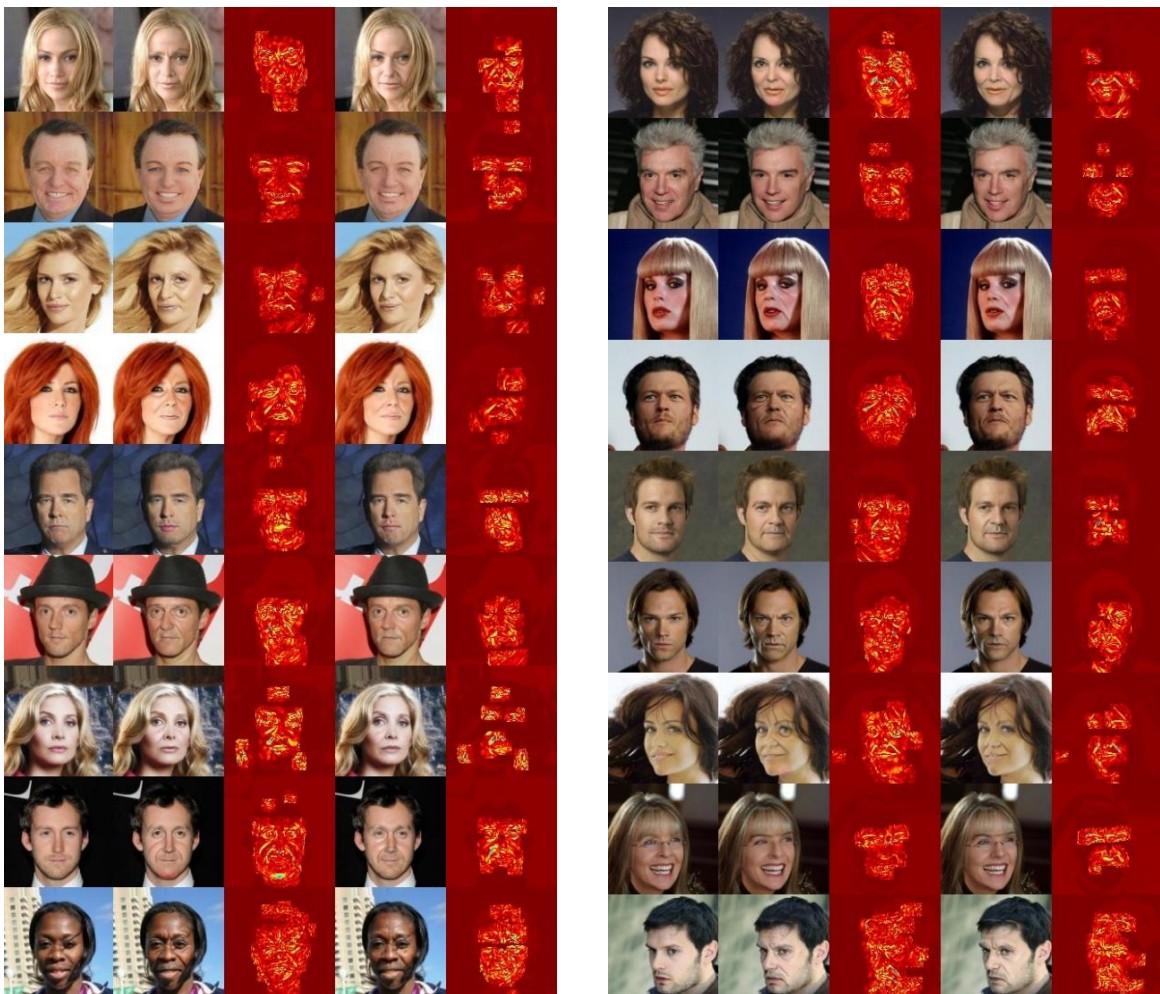

Figure 13: Qualitative comparison on CelebA for the *young* attribute. From left to right: (1) Original image, (2) Baseline method result, (3) Difference map between baseline and original, (4) Our method result, (5) Difference map between our method and original. Our method produces more consistent age transformations while maintaining facial identity.

# E    Stereotype Bias Reduction: Qualitative Analysis

This section provides qualitative evidence demonstrating how our method reduces stereotype bias in counterfactual generation through causal disentanglement and mask-guided intervention.

## E.1    Mechanisms for Stereotype Reduction

Stereotypes in generative models arise when spurious correlations in training data are treated as causal relationships. For example, if "Smiling" and "Female" are correlated in the dataset (but not causally related), a naive counterfactual method might inadvertently feminize a face when adding a smile. Our method addresses this through three mechanisms:

**1. Causal Disentanglement via PAG.** The learned Partial Ancestral Graph (PAG) distinguishes between direct causal relationships and spurious correlations:

- **Direct causal edges (→):** Indicate attributes that should be modified together (e.g., Gray_Hair → Age).

- **Bidirectional edges (↔):** Signal confounding due to latent variables—these regions are *excluded* from modification to prevent stereotype propagation.

**2. Mask-Guided Intervention.** We modify only regions corresponding to direct causal parents $DC(Q_l)$ of the target attribute. Regions associated with confounded or correlated (but non-causal) attributes are preserved, preventing unintended changes that reflect dataset biases rather than true causal effects.

**3. Quantitative Validation.** Table 4 in the main paper shows that our method achieves improved fairness metrics (higher DI, lower SPD) compared to baselines, confirming that causal constraints reduce bias in practice.

### E.2 Case Study: Gender Attribute

Figure 14 illustrates how our method handles the sensitive "Gender" attribute compared to the ACE baseline.

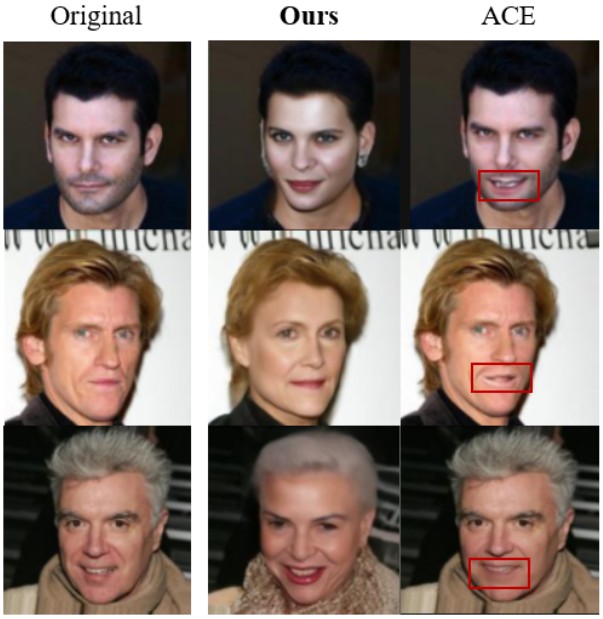

Figure 14: Qualitative comparison for the *Gender* attribute. Red boxes highlight alterations in facial expressions affecting classifier decisions. Row 3: ACE fails to change the classification, while our method succeeds by focusing on causally relevant regions. Our approach avoids modifying spuriously correlated attributes (e.g., "mouth opening"), producing more meaningful and less biased explanations for sensitive attributes.

**Observations:**

- **Baseline (ACE):** The classifier's judgment is influenced by irrelevant facial attributes such as "opening mouth." In the third row, ACE fails to alter the classification result because it modifies regions that are correlated with but not causally related to gender.

- **Ours:** By using the learned PAG to identify direct causes of the "Gender" attribute, our method modifies only causally relevant regions. This produces counterfactuals that successfully change the class while avoiding stereotype-reinforcing modifications.

**Why This Matters for Fairness:** When generating counterfactual explanations for sensitive attributes like gender, it is crucial to avoid reinforcing societal stereotypes. If a method associates "smiling" with "female" (a spurious correlation in many datasets), it may inadvertently:

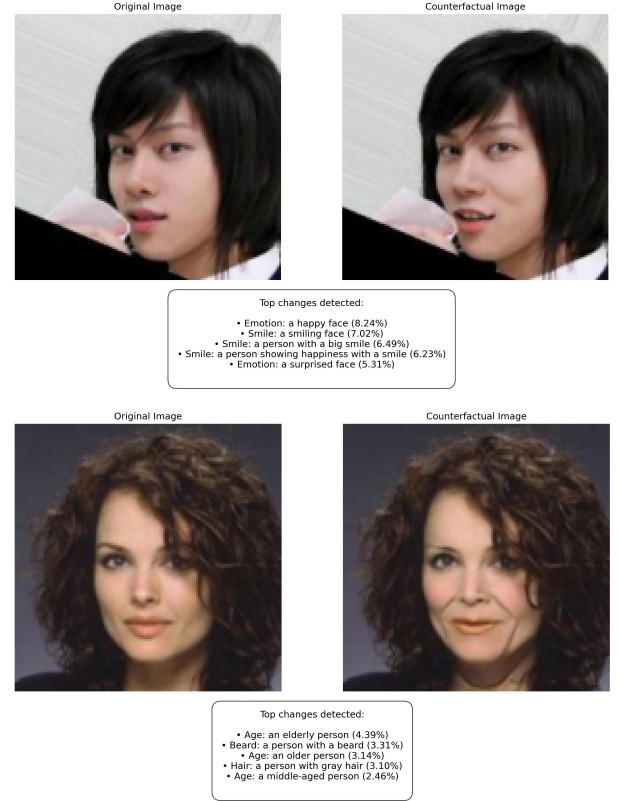

Figure 15: Examples of CLIP-based similarity scores between facial images and descriptive text expressions before and after counterfactual modifications ("Smile": above, "Age": below). Higher similarity scores for relevant facial expressions in the modified images demonstrate the targeted nature of our transformations.

1. Add smiles when changing gender to female, or

2. Remove smiles when changing gender to male.

Such behavior reflects dataset bias, not causal relationships. Our PAG-based approach identifies that Smiling ↔ Gender (bidirectional edge indicating confounding), and therefore excludes smile-related regions from the gender intervention mask. This ensures that gender counterfactuals modify only genuinely gender-related features, producing fairer and more interpretable explanations.

### E.3  Case Study: Age Attribute

Similar principles apply to the "Age" attribute (see Figure in the main paper):

- **Stereotype example:** Baselines often add "Double Chin" when aging a face, reflecting a correlation in the training data.

- **Our approach:** The PAG shows Double_Chin ↔ Age (bidirectional edge), indicating confounding rather than direct causation. We therefore preserve the chin region and modify only direct causes such as hair color (Gray_Hair → Age).

This causal filtering prevents the propagation of age-related stereotypes while still producing effective counterfactual explanations.

### E.4  Summary

Our method reduces stereotype bias by:

1. Learning the causal structure among facial attributes via FCI

2. Identifying and excluding confounded relationships (bidirectional edges)

3. Restricting modifications to direct causal parents only

4. Validating improvements through fairness metrics (DI, SPD) and qualitative inspection

This principled approach ensures that counterfactual explanations reflect true causal relationships rather than spurious correlations that encode societal biases.

# F    Instances of CE in Language via CLIP

To extend the usability of our counterfactual explanations, we employ CLIP to quantify semantic alignment between images and descriptive text. This approach allows us to assess whether the changes introduced by our method are perceptible and semantically meaningful by comparing embedding similarities. Figure 15 presents examples of CLIP-based textual alignment scores before and after counterfactual modifications. These visual comparisons illustrate how our approach effectively alters specific facial attributes while maintaining overall image coherence.

# G    Contribution to Downstream Task

In the synthesis training data perspective. (1) The increased realism of these counterfactuals generated by FaCER helps train more robust and accurate classification models. (2) Realistic counterfactuals provide meaningful variations of the data, helpful in causal inference scenarios that align better with the true underlying distribution, thereby enhancing the generalization capability of the classifier Paulin & Ivasic-Kos (2023). The debiased results in the classification are also discussed. Diverse yet realistic training sets can improve classification by providing more exposure to and knowledge of underrepresented groups in the real world.

