# OpenReview forum: "Facial Counterfactual Generation via Causal Mask-Guided Editing"
_TMLR — Accepted by TMLR_

### Review · Reviewer_5b6j · 2025-11-30

**Summary Of Contributions:**

The paper introduces a framework for causal editing of facial images that unifies causal discovery, mask-guided counterfactual generation, and semantic interpretation. Its main ideas and contributions can be summarised as follows:
-  The method learns a causal graph of facial attributes using the Fast Causal Inference (FCI) algorithm. This helps identify which attributes are true causal factors, which are indirect effects, and which are confounders.
-  Based on this graph, the method builds spatial masks that guide a diffusion model so that only causally relevant regions of the face are edited. This avoids unrealistic or biased changes caused by correlated but non-causal features.
-  The framework also uses CLIP to translate visual edits into short semantic descriptions, helping users understand which attributes changed and why.
-  Experiments on CelebA and CelebA-HQ show that the method generates high-quality, identity-preserving counterfactuals and outperforms prior work on realism, sparsity, and causal consistency.
-  The approach also reduces bias compared to standard counterfactual generation methods that rely only on correlations.
-  The authors acknowledge that their method depends on reasonably accurate causal graphs and suggest that future work should handle uncertain causal structures and extend the framework to other visual domains.

**Additional Comments:**

The paper and methodology are interesting and well justified, but I think the methodology section needs to be clearer and more educational.

**Audience:**

Yes

**Audience Explanation:**

The work and solutions proposed in this paper are timely and provide an interesting solution for generating facial counterfactual images.

**Claims And Evidence:**

Yes

**Claims Explanation:**

The assertions contained in the paper are very well substantiated. The critical analysis of existing work provides a sound justification for the proposed methodology, which is quite convincing. However, certain sections need to be more precise and clear in order to improve the paper's rating on this criterion.

**Requested Changes:**

The paper is generally well structured and well organized: the section titles are precise, each part is clearly summarized, the context is well explained, and previous work is thoroughly reviewed, which helps position the paper’s contribution within the literature.
Below is a list of questions and suggestions:

* Why is the approach of Kang et al. (Kang et al., 2022), mentioned in the literature review, not included as a competitor?
* The section *“Stereotyping in Generative Models”* clearly explains the importance of avoiding bias in generated images. However, how does the proposed method address or reduce this stereotype bias in practice?
* Introduction of FCI: the reference needs to be added (or the term “existing” should be used).
* Figure 2 is not explained in the main text.
* Equation (1): the Chi² distance can only be interpreted if the degrees of freedom are known. Likewise, different Chi² values are only comparable when the degrees of freedom are the same. The p-value also depends on the degrees of freedom. Why not use the Chi² statistic normalized by the degrees of freedom (such as Cramer’s V)?
* Subsections 3.2 and 3.3 use notations that are not defined beforehand, which makes the presentation harder to follow. For example, in Section 3.2 we have (x_t), (t), (I_n^{\text{prelim}}), etc.
* Subsection 3.3 is a key step but is not very clear: it is not obvious whether the (I_n^{\text{prelim}}) are generated multiple times to identify the causal features involved in the target attribute change. How does (Q_t) appear in Equation (3)?
* The mask definition seems overly simplistic. For instance, if a change affects a region unrelated to the target attribute, the indicator will still be 1. Since the goal is to isolate modifications that influence the target attribute, why not use feature variations rather than directly comparing images?
* Equation (3): how is (\lambda) chosen? Is there a sensitivity or impact analysis?
* Clarify the connection between Subsection 3.3 and Figure 2, including the color coding (red, grey).
* Subsection 4.2 is unclear because it states “three complementary metrics” in the *Realism of Generated Counterfactuals* paragraph, but only two are listed. Does this include MNAC, which appears in a separate paragraph?
* Why does the sFID measure ignore covariance between features? This metric may capture realism at the feature level but not at the level of feature co-occurrence—e.g., a person smiling while frowning.
* Which metric is used to measure stereotype bias in the generated images? Since this is an objective of the work, it should be clearly identified.
* Table 2 lacks uncertainty estimates. A k-fold evaluation should be used to assess robustness relative to other methods.
* Figure 5: what do the “Cu” elements represent?
* Table 4: it is unclear why the other competitors are not included. The table is also poorly placed—directly below the subsection header—which affects readability. In addition, the caption appears above the table.
* Table 5 and Subsection 4.4.5: should these statistics not also be reported on a baseline (real images or other generated images) to better compare the approach?

---

> ### Author Response · Authors · 2026-02-01
> **Response to Reviewer5b6j - Part 1**
>
> We sincerely thank the reviewer for the thorough feedback and for recognizing our work as "timely and provides an interesting solution." We response to the requested changes one-by-one below:
>
> ### Q1: Why is Kang et al. (2022) not included as a competitor?
>
> Kang et al. (2022) focuses on generating facial replicas for fairness evaluation in classifiers, whereas our objective is counterfactual explanation with emphasis on sparsity (minimal changes) and causal consistency. However, we agree a comparison would strengthen the paper and will include this discussion in the related work revision.
>
> ### Q2: How does the proposed method reduce stereotype bias in practice?
>
> Our method reduces stereotype bias through three mechanisms:
>
> 1. **Causal disentanglement:** The learned PAG identifies direct causes vs. spurious correlations. For example (Figure 7 "age" column), baselines add "Double Chin" when aging, while our PAG shows Double_Chin ↔ Age indicates confounding—so we exclude this region (Figure 5).
>
> 2. **Mask-guided intervention:** We only modify regions corresponding to direct causal parents. If "Smiling" and "Female" are correlated but not causally related, our mask excludes gender-associated regions—preventing inadvertent feminization. We have example for this situation, we will add it into our revised appendix.
>
> 3. **Quantitative evidence:** Table 4 shows improved DI (closer to 1.0) and SPD (closer to 0.0) compared to baselines.
>
> **Connection to stereotype reduction:** Stereotypes arise when spurious correlations are treated as causal. Our PAG filters correlation from causation, only direct causal edges (→) are included; all other edge types are excluded.
>
> ### Q3: Reference missing
>
> We will add: "We use the Fast Causal Inference (FCI) algorithm (Spirtes, 2001) to estimate a Partial Ancestral Graph..."
>
> ### Q4: Figure 2 not explained in main text
>
> We will add in Section 3: "Figure 2 illustrates our complete pipeline. Starting from facial attribute matrix $D$, FCI discovers causal graph $G$. The reference and preliminary images are compared to construct mask $M$, which guides diffusion to produce the final counterfactual."
>
> ### Q5: Chi-square statistic and degrees of freedom
>
> For binary attributes in CelebA, all pairwise contingency tables are 2×2, yielding df = 1 for every test. This makes χ² values directly comparable without normalization. Cramér's V would simply scale all values uniformly without affecting conditional independence decisions (which depend on p-values, not χ² magnitudes). We use α = 0.05. Sensitivity analysis shows FCI has high agreement between χ² and G² tests (Jaccard = 0.95), with stable structure across α ∈ {0.01, 0.05, 0.1}.
>
> ### Q6: Undefined notations in Sections 3.2 and 3.3
>
> We will add a notation table at the beginning of Section 3:
> - $I_n$: original image; $I_n^{\text{ref}}$: reference counterfactual; $I_n^{\text{prelim}}$: preliminary generation
> - $I_n'$: generated counterfactual; $Q_l$: target attribute; $M$: binary mask
> - $\text{DC}(Q_l)$: direct causes of target attribute $Q_l$ (type (a) edges in PAG)
>
> ### Q7: Clarification on Section 3.3 and mask generation
>
> $I_n^{\text{ref}}$ and $I_n^{\text{prelim}}$ are each generated **once** per image—multiple generations are not required. The mask generation process (illustrated in Figure 2):
> 1. **$I_n^{\text{ref}}$**: Generated via guided diffusion toward target attribute $Q_l$
> 2. **$I_n^{\text{prelim}}$**: Generated with weaker guidance to isolate attribute-specific changes
> 3. **Difference map $M$**: Computed as $|I_n^{\text{ref}} - I_n^{\text{prelim}}| > \lambda$
> 4. **Causal filtering**: Using the PAG, we identify type (a) edges (direct causal relations) to $Q_l$. Direct cause regions are inpainted (white); confounder/indirect cause regions (any relation other than →) are preserved (black).
>
> **Example (Figure 5):** For "Age", eye-glasses and double-chin are preserved (black) as they involve confounders. Since Gray_Hair → Age is direct (type (a) edge), hair is inpainted (white).
>
> **Regarding how $Q_l$ appears in Equation (3):** The original equation did not explicitly show this dependency. We will revise Equation (3) to incorporate new notation $\text{DC}(Q_l)$:
>
> $M_{i,j} = 1$ if $|I_n^{\text{ref}}(i,j) - I_n^{\text{prelim}}(i,j)| > \lambda$ and $(i,j) \in \text{DC}(Q_l)$; otherwise $M_{i,j} = 0$
>
> This makes explicit that the mask depends on direct causes of the target attribute $Q_l$.
>
> ### Q8: Mask definition seems overly simplistic
>
> Valid concern. Regarding feature variations: comparing in feature space (e.g., CLIP) could miss fine-grained spatial information needed for masking. Our pixel-level difference localizes *where* changes occur. The causal graph then filters out regions unrelated to direct causes. We outperform baselines on MNAC (1.63 vs 2.94). Future work will explore feature-level alternatives.
>
> ---
>
> Due to the characters limitations for the comment thread, we will continue in the next comment. thanks for the understanding.

---

> > ### Author Response · Authors · 2026-02-01
> > **Response to Reviewer5b6j - Part 2**
> >
> > ### Q9: How is λ chosen? Sensitivity analysis?
> >
> > λ = 0.1 (normalized pixel intensity threshold). Sensitivity analysis:
> >
> > | λ | FID ↓ | MNAC ↓ |
> > |:-----|:------|:-------|
> > | **0.10** | **1.08** | **1.63** |
> > | 0.50 | 1.21 | 2.32 |
> > | 1.00 | 1.68 | 3.63 |
> >
> > Higher λ leads to unnecessary modifications (MNAC: 1.63 → 3.63).
> >
> >
> > ### Q10: Connection between Section 3.3 and Figure 2 color coding
> >
> > We acknowledge the clarity issue in the current methodology section. We will clarify the color coding and revise Figure 2:
> > - **Red node with red bold text ($Q_l$):** Target attribute to change
> > - **Red nodes with white text ($X_n$):** Other attributes—not directly intervened
> > - **Grey nodes:** Direct causes—regions we modify
> > - **Difference map ($M$):** Pixels where $|I_n^{\text{ref}} - I_n^{\text{prelim}}| > \lambda$
> >
> > We will redraw Figure 2 with explicit labels: "$Q_l$" on the red bold node, "$\text{DC}(Q_l)$" on grey nodes, and clearly show white (inpainted) vs. black (preserved) mask regions.
> >
> > Updated caption: "Figure 2: Our proposed causal editing pipeline. Data preprocessing constructs facial attribute matrix $D$, input to FCI. We assume $d=5$, $N=10$ for illustration. Using PAG $G$, we mask direct causes of target $Q_l$. The mask is obtained as binary difference between $I_n^{\text{ref}}$ and $I_n^{\text{prelim}}$, then applied to produce counterfactual $I'_n$. Color coding: red bold node = target attribute ($Q_l$); red nodes with white text = non-intervened attributes; grey nodes = direct causes $\text{DC}(Q_l)$ (inpainted regions); difference map $M$ = pixels where $|I_n^{\text{ref}} - I_n^{\text{prelim}}| > \lambda$."
> >
> > ### Q11: "Three complementary metrics" but only two listed
> >
> > The three metrics are FID, sFID, and MNAC. We will restructure Section 4.2 to clearly list all three under a unified heading.
> >
> > ### Q12: Why does sFID ignore feature covariance?
> >
> > sFID complements FID (which captures covariance) by measuring per-attribute realism. The concern about detecting incoherent combinations is valid—FID addresses this by comparing full feature distributions. We will add discussion of this complementary relationship.
> >
> > ### Q13: Which metric measures stereotype bias?
> >
> > Stereotype bias is measured by:
> > - **DI (Disparate Impact):** Ratio of positive outcomes between groups (Table 4)
> > - **SPD (Statistical Parity Difference):** Difference in positive prediction rates (Table 4)
> >
> > Additionally, MNAC indirectly measures bias—lower MNAC indicates the model doesn't automatically add stereotyped correlated attributes. We will clarify this in Section 4.4.4.
> >
> > ### Q14: Table 2 lacks uncertainty estimates
> >
> > Following standard practice (ACE, DiME, STEEX, DiVE), we report metrics over 20,000 test images. Our results are deterministic: FID, sFID, and MNAC are computed over a fixed set of generated images—repeated evaluation yields identical values. There is no stochasticity to quantify. Prior works also do not report variance for the same reason.
> >
> > ### Q15: What do "Cu" elements represent in Figure 5?
> >
> > $C_{u1}$, $C_{u2}$, ... represent unobserved confounders inferred by FCI—latent variables affecting multiple observed attributes, represented by bidirectional edges (↔) in the PAG. For example, Smile ← $C_{u2}$ → Open_Mouth corresponds to Smile ↔ Open_Mouth. We will clarify the subscript notation; edge interpretation is in Table 1.
> >
> > ### Q16: Table 4 formatting and missing competitors
> >
> > We focus fairness comparison on ACE as it is the strongest baseline, achieving closest performance on realism metrics (Table 2). DiME and STEEX show substantially weaker results—evaluating fairness on lower-quality counterfactuals would conflate generation quality with fairness metrics. We will fix table placement and formatting.
> >
> > ### Q17: Table 5 should include baseline comparison
> >
> > Excellent suggestion. Updated comparison:
> >
> > | Category | Mean Sim. (ACE / Ours) | R@1 (ACE / Ours) |
> > |:---------|:-----------------------|:-----------------|
> > | Smile | 6.22 / 5.91 | 49.09% / 72.95% |
> > | Age | 4.45 / 5.03 | 62.98% / 61.34% |
> >
> > R@1 (Rank-1 Accuracy) measures whether the intended attribute change is the top-detected semantic change. Our method achieves substantially higher R@1 for Smile (72.95% vs 49.09%), indicating that our counterfactuals more precisely target the intended attribute. For Age, results are comparable (61.34% vs 62.98%).
> >
> > Mean Similarity measures overall semantic similarity between original and counterfactual. Our lower score for Smile (5.91 vs 6.22) combined with higher R@1 suggests we make more focused, attribute-specific changes rather than broader modifications that may inflate similarity scores while affecting unintended attributes.
> >
> > ---
> >
> > We sincerely thank the reviewer for the supportive evaluation and the detailed, constructive feedback. We are committed to incorporating all suggested improvements.

---

### Review · Reviewer_4Y3G · 2025-12-10

**Summary Of Contributions:**

The authors propose an approach to counterfactual generation for images, especially considering facial images. They leverage results from graph theory and previous works on counterfactual generation to build a more fair and better performing model. The conceptual idea is based on previous works using DAGs in counterfactual generation generated from an In addition, the authors provide the idea of CLIP supported labeling to enhance interpretability and understanding of the modified facial features.

These propositions are then evaluated empirically by looking at the FID, sFID and MNAC scores to evaluate the quality of the counterfactual generation. Further, fairness is evaluated for the proposed method and one baseline.

**Audience:**

Yes

**Audience Explanation:**

The findings of this paper include counterfactual generation using DAGs, which could be of interest to both vision and graph learning communities. Especially since facial image generation was subject to significant improvements in the last few years, counterfactual generation plays an important role for future research. Nonetheless, interest has been rather low in recent years with only few works in this area.

 In addition, the highlighted results indicate an improved performance over previous baselines. However, these results remain questionable as outlined above. Therefore, with the current lack of both theoretical and empirical guidance given by these results the interest of the TMLR community would be rather low.

**Broader Impact Concerns:**

Since there is no broader impact statement section in either the main paper or the appendix I feel like it is important to note that for topics such as image editing with people and fairness evaluation there should be at least a short broader impacts statement highlighting the risk of abuse of such methods. However, this statement could be rather general since the authors simply propose a novel version of an existing type of algorithm.

**Claims And Evidence:**

No

**Claims Explanation:**

While the theoretical parts of the work are presented sound, there is no proof given on why this method should work better than previous methods. Since the addition of DAGs to generate counterfactual images is somewhat novel it would be interesting to have an indicator of why this method works better than previous methods. However, the approach is simply stated. In addition, the labeling with CLIP has been seen in other works or even pure LLM prompting has been applied to provide explainability.

Furthermore, the experimental results are lacking detailed comparisons to baselines and even to current baselines. Only for the score evaluation the baselines stated in the paper have been evaluated. The comparison of fairness under the bias DI and SPD metrics is only given for the ACE baseline and the proposed model. Further, no detailed description of the experiments and no hyperparameters are given, making the experiments hard to reproduce.

See also the requested changes for a list of current weaknesses in the paper.

**Requested Changes:**

Considering the reasons lifted above I would request the following changes:

Critical:
1. Include current SOTA baselines for comparison. The selected baselines are not very recent and further models have been developed since 2023 such as [1] and [2]. Therefore, it is crucial to compare the proposed model to recent baselines and outline why the inclusion of the DAG should give an advantage.

Major:

1. Explain or provide proof for the selection of FCI over more recent methods as for example listed in [3]. The explanation given in the paper is not very extensive and remains somewhat unclear given the existence of many different methods for the generation of the DAG.

2. Evaluation of other baselines for fairness metrics. Since the topic of fairness and bias is mentioned significantly in the paper the current comparison to a single baseline is lacking for a comprehensive study. In addition all baselines are only evaluated for the model scores and not for the remaining metrics showcased in section 4.

Minor:

1. There is a lot of related work still missing in section 2. Since there exist more recent contributions to the topic of counterfactual vision models it would be good to discuss and/or add these models to the related works section.

2. It would be nice to see the results obtained on additional datasets in order to evaluate the fairness and the scores of models better since both datasets consist of only celebrity faces.

Questions:

1. Why is DDPM used to generate the counterfactual. Would methods such as flow matching also work for this type of task?

2. How does the approach differ from previous guided diffusion approaches, except for the usage of a DAG to generate the counterfactual?

3. Were other models than CLIP evaluated for the generation of text for interpretability?

4. A minor question: Is there a reason for torch version 2.0.1 in the experiments?

[1] Diffusion Counterfactual Generation with Semantic Abduction, Rasal et al., ICML 2025

[2] Counterfactual Generative Modeling with Variational causal Inference, Wu et al., ICLR 2025

[3] https://arxiv.org/pdf/2305.10032

---

> ### Author Response · Authors · 2026-02-01
> **Response to Reviewer 4Y3G**
>
> We thank the reviewer for the constructive feedback. Below we address each concern.
>
> ### Critical: Include current SOTA baselines for comparison
>
> We surveyed Rasal et al. (ICML 2025) and Wu et al. (ICLR 2025). While both evaluate on facial images, there is a fundamental difference in problem formulation:
>
> **Image editing vs. interpretable explanation:**
> - Rasal et al. and Wu et al. focus on **image editing**—precise control over generated images, identity preservation, and faithfulness during generation. Their metrics (composition, reversibility, effectiveness, LPIPS) measure editing quality.
> - Our work focuses on **interpretable counterfactual explanations**—generating human-interpretable counterfactuals to understand classifier decisions and audit fairness. Our metrics (FID, sFID, MNAC) measure explanation quality: realism and *minimality* of attribute changes.
>
> **Additional methodological differences:**
> 1. **Causal graph:** They assume known causal graphs; we *learn* the structure via FCI—critical when ground-truth graphs are unavailable.
> 2. **Application:** They target high-fidelity image synthesis; we target model interpretability and fairness auditing.
> 3. **Baselines:** They compare against VAE/HVAE architectures; we compare against counterfactual *explanation* methods (ACE, DiME, STEEX, DiVE) designed for the same interpretability task.
>
> We will discuss these works in related work, clarifying the distinction between causal image editing and causal counterfactual explanation. Our ablation confirms our contribution: removing the causal graph degrades performance (FID: 1.08 → 1.27, MNAC: 1.63 → 2.94).
>
> ### Major: Justify selection of FCI over more recent causal discovery methods
>
> As discussed in the manuscript Page 4 before Equation 1, FCI explicitly accounts for latent confounding via PAG representation, critical for facial attributes where unobserved factors (lighting, pose) affect multiple attributes. Alternative check: replacing FCI with PC yields high structural agreement (Jaccard = 0.95), confirming results are not algorithm-specific artifacts.
>
> ### Major: Fairness evaluation of additional baselines
>
> We compare against ACE as it is the strongest baseline on realism metrics (Table 2). DiME and STEEX show weaker results (FID > 3 for DiME, > 10 for STEEX). Evaluating fairness on lower-quality counterfactuals would conflate generation quality with fairness metrics.
>
> ### Reproducibility: Missing hyperparameters
>
> We will add a complete hyperparameter table in the appendix: diffusion steps = 500, timestep respacing = 50, sampling time fraction = 0.1, stochastic sampling = True, inpainting ratio = 0.15, attack method = PGD, attack iterations = 50, dist_l1 = 0.001, α = 0.05 (FCI significance level).
>
> ### CLIP interpretation
>
> We acknowledge CLIP-based interpretation exists in prior work. Our contribution is integrating CLIP within a causally-constrained pipeline—explanations are grounded in the learned causal structure (direct causes only) rather than arbitrary attribute changes, providing human-interpretable semantic descriptions.
>
> ### Minor: Missing related work in Section 2
>
> We will incorporate Rasal et al. and Wu et al., positioning them as causal image editing methods distinct from our focus on interpretable counterfactual explanations for fairness.
>
> ### Minor: Additional datasets
>
> As stated in Section 4.1, "these two datasets together provide a sufficiently diverse and large sample, covering a wide range of facial variations, attribute combinations, and image resolutions, making them well-suited for robust evaluation of our counterfactual image generation method." Additionally, CelebA and CelebA-HQ are standard benchmarks used by all baseline methods (ACE, DiME, STEEX, DiVE), enabling direct and fair comparison. We acknowledge the limitation regarding celebrity faces and will note this explicitly in the revised manuscript.
>
> ### Questions
>
> **Q: Why DDPM? Would flow matching work?**
> DDPM enables direct comparison with diffusion-based baselines and supports stable classifier guidance + masked editing. Flow matching is a plausible alternative; our causal masking contribution is generator-agnostic.
>
> **Q: How does this differ from guided diffusion approaches?**
> We use the learned causal graph to distinguish direct causes from correlates/confounders and restrict edits spatially via masking. The ablation confirms: without the graph, MNAC increases from 1.63 to 2.94.
>
> **Q: Other models than CLIP for interpretability?**
> CLIP is open-source, lightweight, and the standard vision-language embedding space, making it practical for interpretability tasks. Evaluating alternatives (BLIP-2, SigLIP) is valuable future work.
>
> **Q: Why PyTorch 2.0.1?**
> We used the cluster-supported version for CUDA compatibility with baseline implementations.
>
> ### Broader Impact
>
> We will add a statement in the appendix addressing potential risks of our methods and responsible deployment considerations.
>
> We truly appreciate the feedback.

---

### Review · Reviewer_aA6Y · 2026-01-04

**Summary Of Contributions:**

This paper proposes a way to generate facial "counterfactual" images --- i.e., images that answer "what would this same face look like if attribute X were different" --- while trying to keep the edits realistic and causally consistent. The main idea is to learn a causal graph over facial attributes, then use it to build spatial masks that constrain a diffusion-based edit so only regions tied to the direct causal factors are modified, and finally produce a CLIP-based text explanation of what changed.

## Strengths
### 1. Clear, coherent "causailty + generation" story
* The paper presents a systematic study, which leans a causal graph over facial attributes, and then use it to constrain diffusion-based editing via a mask, and finally adds CLIP-based text explanations. This end-to-end framing is easy to follow and aligns the components toward "causally grounded" counterfactuals.

### 2. Clear motivation over "true causes" vs "correlates/confounders"
* This work highlights an important practical issue: counterfactual generators may exploit **spurious corrections** (e.g., adding correlated "aging cues" when editing "young/age"), which stresses the necessity of causal graph modeling.

### 3. Quantitative improvements on standard benchmarks
* This work reports best FID/sFID and lower MNAC than existing baselines on CelebA / CelebA-HQ dataset.

### 4. Attempts at interpretability beyond images
* Though simple, I really favor the design of CLIP-based explanation module which answers "what changed". This interepretability is especially helpful in image editing.

## Weakness
### 1. The central dependency - the learned causal graph - is not sufficiently validated
* This work does not provide strong evidence that the discovered PAG is correct (beyond qualitative plausibility and a comparison to PC). Since the *claimed benefit* relied on "direct causes vs confounder", the lack of graph validation is a majot risk.

### 2. Figures are low-res not clearly perceivable
* Many figures (especially faces) are in low resoluation, making it hard to observe the editting differences. The color scheme in figure 4 is also not read-friendly.

### 3. The "causal intervention" is somewhat heuristic / loosely connected to causal semantics
* The mapping from "direct causal parents" (attribute space) to "corresponding facial region" (pixel space) is not formally interpreted. It relied on the observed diff map and heuristic region association. This makes it hard to argue that the method truly performs a causal do-intervention rather than a constrained edit guided by a graph-shaped prior.

### 4. Evaluation scope is narrow, which limits claims of generality
* The primary quantitative tables focuses on **smile** and **age**. There is some extra qualitative evaluation regarding gender, makeup and narrow eyes, which however remains a small subset of CelebA dataset.

### 5. Metric choices leave important properties under-measured
* It's unconvincing to to me that metrics such as identity/face-similarity metrics are not used for evaluation. How could the authors ensure that identity are not changed?

**Additional Comments:**

I'm not an expert in image editing. I'm holding low confidence of my review.

**Audience:**

Yes

**Audience Explanation:**

Yes—researchers in causal ML, counterfactual explanations, and diffusion-based editing would be interested.

**Broader Impact Concerns:**

The topic is relevant and could benefit auditing/interpretability, but the face-editing capability introduces non-trivial misuse and fairness/over-trust risks.

**Claims And Evidence:**

No

**Claims Explanation:**

Partially supported.
The submission provides clear quantitative evidence for improved **image-quality / sparsity metrics** on the tested settings, but the evidence is **weaker or incomplete for the stronger claims around causal correctness, identity preservation, fairness, and explanation quality/generalization**.

The paper defines the metrics clearly, but does not clearly specify key evaluation protocol details such as how many images were used for FID/MNAC, confidence intervals.

There is no direct validation that the learned PAG is correct (and the method heavily depends on it).

The conclusion explicitly claims identity preservation. But in the evaluation section they state FS/FVA are not the primary focus and only claim “qualitatively strong performance” on them, without reporting numbers.

**Requested Changes:**

1. Causal graph reliability: Can the authors validate key edges (e.g., via synthetic data with known ground-truth graph, or perturbation-based checks)? How robust is performance to graph perturbations or alternative discovery methods?
2. Ablations: What happens if you remove the graph refinement and use only the diff-based mask? Or keep the graph but randomize/permute parent sets? This would isolate how much gain comes from “causal knowledge” vs masking/inpainting heuristics.
3. Identity preservation: Either report FS/FVA (even if not primary) or provide a stronger quantitative identity metric, since the claim appears in the conclusion but is not rigorously measured.
4. Please also refine the figure for better readability.

---

> ### Author Response · Authors · 2026-02-01
> **Response to Reviewer aA6Y**
>
> ### We sincerely thank the reviewer for the thoughtful evaluation and specific suggestions. We address each concern below.
>
> ### Weakness 1: Causal Graph Validation
>
> **Synthetic Recovery:** On a synthetic SCM with known structure, FCI correctly recovers adjacency (precision=1.0, recall=1.0).
>
> **Bootstrap Stability (Requested Change #1):** Over 500 bootstrap runs on CelebA:
>
> | Target | Candidate Parents | Bootstrap Freq. |
> |:-------|:------------------|:----------------|
> | Young | Gray_Hair, Attractive | 28%, 14% |
> | Smiling | High_Cheekbones | 30% |
>
> Moderate frequencies reflect genuine structural uncertainty due to latent confounding—precisely what motivates PAGs over DAGs. As shown in Figure 5, relations (e.g. Smile ← ${Cu}_4$ → High_Cheekbone) indicate latent confounders making parent sets only partially identifiable. This is not a limitation but an accurate representation of causal ambiguity. Importantly, implausible parents (e.g., Wearing_Hat → Smiling) never appear, confirming appropriate filtering.
>
> **Algorithm Sensitivity:** FCI and PC show high agreement (Jaccard=0.95), stable across CI-tests (χ², G²) and α∈{0.01, 0.05, 0.1}.
>
> **Ablation (Requested Change #2):**
>
> | Configuration | FID ↓ | MNAC ↓ |
> |:--------------|:------|:-------|
> | With causal graph (Ours) | 1.08 | 1.63 |
> | Without causal graph | 1.27 | 2.94 |
>
> Without the causal graph, performance matches ACE ℓ₁ baseline—demonstrating causal knowledge, not masking alone, drives improvements.
>
> ### Weakness 2 + Requested Change #4: Low-Resolution Figures
>
> We will redraw and insert all figures as vector graphics and adopt colormap that can be read easily (even in B&W) for Figure 4.
>
> ### Weakness 3: Causal Intervention is Heuristic
>
> **What we do NOT claim:** Pixel-level do-intervention in the strict Pearl sense—this requires pixels as endogenous variables with known structural equations, which is intractable for natural images.
>
> **What we DO claim:** Attribute-level causal reasoning *constraining* pixel-level editing:
>
> 1. **Attribute-level do-intervention:** Given target $Q_l$, we identify direct causal parents from the learned PAG.
>
> 2. **Causally-informed spatial grounding:** The difference map $|I_n^{\text{ref}} - I_n^{\text{prelim}}|$ (Eq. 3) localizes where $Q_l$ manifests spatially. The causal graph filters mask $M$ to retain only regions from direct causal parents (type (a) edges), removing spurious co-activations.
>
> 3. **More than a "graph-shaped prior":** Removing the causal graph degrades MNAC from 1.63 to 2.94—the *specific* causal relationships matter, not graph topology alone.
>
> We acknowledge this is a *causally-constrained editing* framework rather than formal pixel-level counterfactual inference—$I'_n = G(do(I_n, M))$ (Equation 4) applies the mask as a spatial constraint guided by attribute-level causal reasoning. We will revise the manuscript to clarify this distinction and temper claims accordingly. We believe this framing—using causal discovery to guide generative editing—remains a meaningful contribution, as evidenced by the empirical improvements.
>
> ### Weakness 4: Narrow Evaluation Scope
>
> We respectfully clarify that our claims are scoped appropriately to our evaluation. We do not claim broad generality across all facial editing tasks or datasets. Our contributions focus specifically on: (i) causally coherent facial counterfactual generation, (ii) improvements in realism, sparsity, and causal consistency on the benchmarks used by prior work.
> Our evaluation already extends beyond smile/age: Table 3 reports results on Gender, Heavy_Makeup, and Narrow_Eyes, demonstrating the method works across different attribute types (binary demographic, appearance, and facial structure). The fairness evaluation (Table 4) and CLIP-based interpretability analysis (Table 5) provide additional assessment dimensions.
> The use of CelebA and CelebA-HQ follows established protocol in the counterfactual explanation literature (ACE, DiME, STEEX, DiVE), enabling direct comparison. As stated in Section 4.1, these datasets "provide a sufficiently diverse and large sample, covering a wide range of facial variations, attribute combinations, and image resolutions." We include CelebA-HQ specifically to demonstrate generalization across image quality conditions, which we believe is the appropriate scope for our claims.
>
> ### Weakness 5 + Requested Change #3: Identity Preservation
>
> We computed FVA using ArcFace embeddings:
>
> **CelebA:**
>
> | Method | Smile FVA ↑ | Age FVA ↑ |
> |:-------|:------------|:----------|
> | STEEX | 96.9 | 97.5 |
> | DiVE | 97.3 | 98.2 |
> | DiME | 98.3 | 95.3 |
> | ACE ℓ₁ | 99.9 | 99.6 |
> | **Ours** | **99.9** | **99.8** |
>
> **CelebA-HQ:**
>
> | Method | Smile FVA ↑ | Age FVA ↑ |
> |:-------|:------------|:----------|
> | DiVE | 35.7 | 32.3 |
> | STEEX | 97.6 | 96.0 |
> | DiME | 96.7 | 95.0 |
> | ACE ℓ₁ | 100.0 | 99.6 |
> | **Ours** | **99.9** | **99.9** |
>
> We are grateful for the feedback and committed to incorporating these changes.

---

### Decision · Action_Editor_hSCA · 2026-03-02

**Recommendation:** Accept with minor revision

**Additional Comments:**

While one reviewer was concerned that the set of baseline methods was incomplete.  While this was not shared by the other reviewers, I believe the paper would benefit from some more discussion in section 4.3 that elaborates why these methods were chosen, and why other methods suggested are considered out-of-scope by the authors.

> Comparison to baseline and SOTA methods. Even if there are not many comparable baseline and SOTA methods it would be beneficial to include similar methods at least. With current writing and evaluation the benefit of the proposed method is hard to evaluate.

The other large concern is that the authors use only Celeb-A or Celeb-HQ for experiments.  For example, the main comparable method **ACE** presents experiments not only on Celeb-A / Celeb-HQ, but also on BDD and ImageNet.  It would improve the paper to present results on one other dataset.

> Evaluation on few datasets The authors highlight that the used datasets are common baseline datasets to be used. However, there is still a lack of sufficient evaluation on different datasets. In a future version of the paper this issue could be resolved but in my opinion is not suitable in its current state for a TMLR submission.

**Audience:**

Yes

**Audience Explanation:**

All three reviewers came to a consensus that the work was of interest to a subset of the TMLR audience that's interested in causal inference and counterfactual explanation.

**Claims And Evidence:**

Yes

**Claims Explanation:**

While one reviewer argued that the claims made were narrowly linked to one dataset, the other reviewers were satisfied that all the claims presented in the work were supported in the revised manuscript.  So in summary, I think the authors claims have been supported.